# Cell-autonomous regulation of epithelial cell quiescence by calcium channel Trpv6

Yi Xin, Allison Malick, Meiqin Hu, Chengdong Liu, Heya Batah, Haoxing Xu, Cunming Duan*

Department of Molecular, Cellular and Developmental Biology, University of Michigan, Ann Arbor, United States

**Abstract** Epithelial homeostasis and regeneration require a pool of quiescent cells. How the quiescent cells are established and maintained is poorly understood. Here, we report that Trpv6, a cation channel responsible for epithelial $Ca^{2+}$ absorption, functions as a key regulator of cellular quiescence. Genetic deletion and pharmacological blockade of Trpv6 promoted zebrafish epithelial cells to exit from quiescence and re-enter the cell cycle. Reintroducing Trpv6, but not its channel dead mutant, restored the quiescent state. $Ca^{2+}$ imaging showed that Trpv6 is constitutively open in vivo. Mechanistically, Trpv6-mediated $Ca^{2+}$ influx maintained the quiescent state by suppressing insulin-like growth factor (IGF)-mediated Akt-Tor and Erk signaling. In zebrafish epithelia and human colon carcinoma cells, Trpv6/TRPV6 elevated intracellular $Ca^{2+}$ levels and activated PP2A, which down-regulated IGF signaling and promoted the quiescent state. Our findings suggest that Trpv6 mediates constitutive $Ca^{2+}$ influx into epithelial cells to continuously suppress growth factor signaling and maintain the quiescent state.

## Introduction

Quiescence is a non-proliferative cellular state found in many cell types in the body. While non-proliferative, these cells retain the ability to re-enter the cell cycle in response to appropriate cell-intrinsic and extrinsic signals (*Matson and Cook, 2017*; *Sun and Buttitta, 2015*; *Yao, 2014*). Quiescence protects long-lived cells, such as adult stem cells against the accumulation of genomic aberrations and stress. Maintaining a pool of quiescent cells is critical for tissue repair, wound healing, and regeneration (*Cheung and Rando, 2013*). This is particularly important for epithelia which are rapidly and continuously renewed throughout life. The intestinal epithelial cells, for example, are renewed every 4 to 5 days (*van der Flier and Clevers, 2009*). By synchronizing cultured mammalian cells in G0 via serum starvation followed by serum re-stimulation, *Yao et al. (2008)* showed that the Rb proteins (pRb, p107, and p130) and their interactions with E2F proteins are critical in regulating the proliferation-quiescence decision (*Yao et al., 2008*). Acting downstream, a bifurcation mechanism controlled by CDK2 activity and p21 regulating the proliferation-quiescence decision has also been demonstrated in cultured mammalian cells (*Spencer et al., 2013*). While important insights have been learnt from in vitro studies, how the quiescent cell pools are established during development and maintained in vivo is not well understood. The exceptionally high turnover rate implies that cell-type-specific mechanism(s) must exist.

The transient receptor potential cation channel subfamily V member 6 (TRPV6) is expressed in mammalian intestinal epithelial cells (*Hoenderop et al., 2005*). TRPV6 is a conserved calcium channel that constitutes the first and rate-limiting step in the transcellular $Ca^{2+}$ transport pathway (*Hoenderop et al., 2005*; *Peng et al., 1999*; *Peng et al., 2000*; *Wissenbach et al., 2001*). In zebrafish, *trpv6* is specifically expressed in a population of epithelial cells known as ionocytes or NaR cells (*Dai et al., 2014*; *Pan et al., 2005*). NaR cells take up $Ca^{2+}$ from the surrounding habitats into the body to maintain body $Ca^{2+}$ homeostasis (*Liao et al., 2009*; *Yan and Hwang, 2019*). NaR cells are

*For correspondence:
cduan@umich.edu

Competing interests: The authors declare that no competing interests exist.

polarized cells that functionally and molecularly similar to human intestinal epithelial cells. While located in the gill filaments and the intestine in the adult stages, these cells are distributed in the yolk sac skin during the embryonic and larval stages, making these easily accessible for experimental observation and perturbations (*Dai et al., 2014*; *Pan et al., 2005*). When zebrafish are grown in homeostatic normal [Ca$^{2+}$] conditions, NaR cells are maintained in a quiescent state and the Akt-Tor activity is regulated at low levels. Low [Ca$^{2+}$] stress increases Akt-Tor activity in these cells and promotes their re-entry into the cell cycle (*Dai et al., 2014*; *Liu et al., 2017*). This is similar to the proposed role of mTOR signaling in adult stem cells (*Kim and Guan, 2019*; *Meng et al., 2018*), suggesting an evolutionarily conserved mechanism(s) at work. More recent studies suggest that insulin-like growth factor binding protein 5a (Igfbp5a), a secreted protein that binds IGF with high-affinity, plays a critical role in activating Akt-Tor signaling in these cells via the IGF1 receptor under calcium-deficient states (*Liu et al., 2018*). The mechanism controlling the quiescent state under normal [Ca$^{2+}$] condition is currently unknown. In a previous study, we found that zebrafish *mus* mutant larvae, a loss-of-function Trpv6 mutant fish line obtained from an ENU mutagenesis screen (*Vanoevelen et al., 2011*), had many proliferating NaR cells and elevated Akt-Tor signaling, suggesting Trpv6 may play a negative role in regulating NaR cell proliferation (*Dai et al., 2014*). How does Trpv6 act to inhibit Akt-Tor signaling and whether it involves in cell quiescence regulation are unknown. Because TRPV6/Trpv6 is the primary Ca$^{2+}$ channel responsible for epithelial Ca$^{2+}$ uptake and since Ca$^{2+}$ is a major second messenger involved in cell proliferation and differentiation in many cell types (*Clapham, 2007*; *Hoenderop et al., 2005*), we hypothesized that Trpv6 regulates the quiescent state by conducting Ca$^{2+}$ influx into epithelial cells and suppressing IGF1-receptor-mediated signaling. The objective of this study was to test this hypothesis and to elucidate the underlying mechanisms of Trpv6 action.

## Results

### Trpv6 is crucial for epithelial Ca$^{2+}$ uptake in zebrafish

Three *trpv6* mutant fish lines were generated using CRISPR/Cas9 (*Figure 1A*). All three Trpv6 mutant proteins lack the six transmembrane domains and the critical ion pore region and are predicted to be null mutations (*Figure 1B*). The *trpv6Δ7* and *trpv6Δ8* lines were made in the *Tg(igfbp5a:GFP)* fish background. *Tg(igfbp5a:GFP)* is a transgenic fish line expressing EGFP in the *trpv6*-expressing NaR cells (*Liu et al., 2017*), allowing real-time analysis of NaR cell proliferation. The *trpv6Δ8–2* line was in a non-transgenic fish background and used in Ca$^{2+}$ imaging analysis described later. The gross morphology and body size of the mutant fish were similar to their siblings (*Figure 1—figure supplement 1*). All mutant fish died within 2 weeks (*Figure 1C and D*). Alizarin red staining indicated a marked reduction in the calcified bone mass in the *trpv6$^{-/-}$* mutant fish (*Figure 1E*), indicating body calcium deficiency. Fura-2 Ca$^{2+}$ imaging experiments in HEK293 cells transfected with zebrafish Trpv6 and human TRPV6 were performed. The Trpv6-mediated [Ca$^{2+}$]$_i$ change was similar to that of TRPV6 (*Figure 1F*). D542 in mammalian TRPV6 occupies a critical position in the ion pore region and mutation of this residue abolishes its Ca$^{2+}$ permeability (*McGoldrick et al., 2018*; *Sakipov et al., 2018*). This residue is conserved in zebrafish Trpv6 at position 539 (*Figure 1—figure supplement 2*). We generated and tested Trpv6D539A mutant. The [Ca$^{2+}$]$_i$ levels in Trpv6D539A mutant transfected cells were low and did not respond to changes in extracellular [Ca$^{2+}$] (*Figure 1F*). The maximal Ca$^{2+}$ influx rate was reduced to a negligible level in Trpv6D539A transfected cells (*Figure 1G*). Whole-cell patch clamp experiments confirmed that the Trpv6 mediated Ca$^{2+}$ current and this activity was abolished in the Trpv6D539A mutant (*Figure 1H*). These findings support the notion that Trpv6 plays an indispensable role in epithelial Ca$^{2+}$ uptake and maintaining body Ca$^{2+}$ balance and provided critical reagents for subsequent experiments.

### Trpv6 regulates the quiescence-proliferation decision in epithelial cells

To determine the possible role of Trpv6 in NaR cells, double-blind tests were performed (*Figure 2A*). In agreement with previous studies (*Dai et al., 2014*; *Liu et al., 2017*), NaR cells in the wild-type and heterozygous siblings were distributed in the yolk sac region as single cells in a salt-and-pepper pattern (*Figure 2B*). NaR cells in the *trpv6Δ8* mutant larvae were often observed in clusters of newly divided cells (*Figure 2B*). These proliferating NaR cells had enlarged apical opening

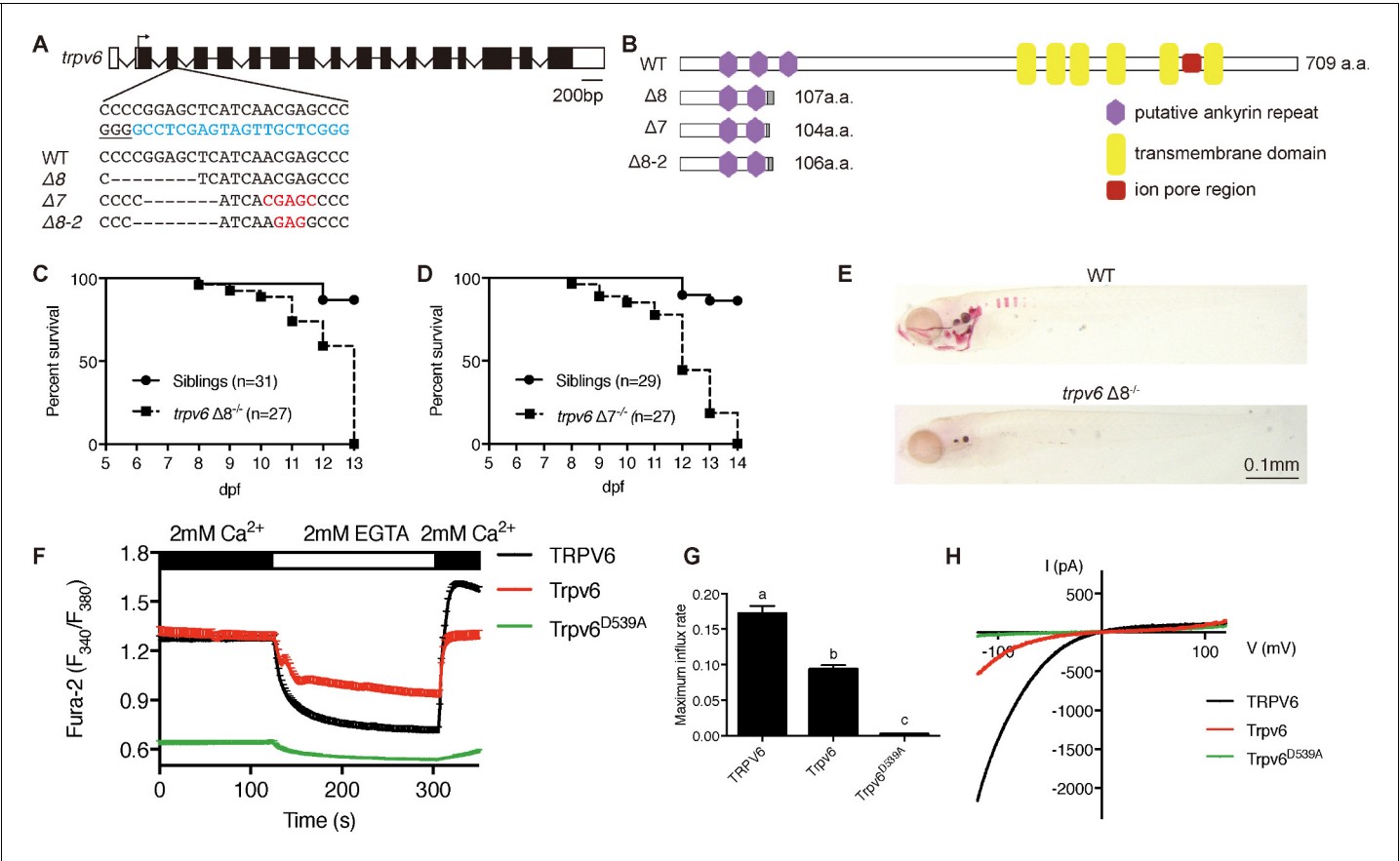

**Figure 1.** Genetic deletion of the conserved epithelial calcium channel Trpv6 results in calcium deficiency and premature death. (A) Schematic diagram showing *trpv6* gene (WT) and various mutant sequences. Filled boxes indicate *trpv6* ORF and open boxes indicate UTRs. Introns are shown as lines. The gRNA targeting site is indicated in blue color and PAM motif is underlined. Deleted and inserted nucleotides are indicated by dash lines and red letters, respectively. (B) Schematic diagram of Trpv (WT) and its mutants. Putative functional domains are indicated. The gray box indicates altered sequence caused by frame shifts. (C–D) Survival curves of *trpv6Δ8⁻/⁻; Tg(igfbp5a:GFP)* (C) and *trpv6Δ7⁻/⁻; Tg (igfbp5a:GFP)* fish (D) and siblings. The numbers of total fish are indicated. (E) Representative images of Alizarin red stained wild-type and *trpv6Δ8⁻/⁻; Tg(igfbp5a:GFP)* fish at 7 days post fertilization (dpf). (F) Fura-2 Ca²⁺ imaging analysis of HEK293 cells transfected with the indicated plasmids. n > 50 cells from three independent experiments. (G) The maximal influx rate. n = 3 independent experiments. (H) Currents evoked by a RAMP voltage from −120 mV to +120 mV in HEK293 cells transfected with the indicated plasmids. In this and all subsequent figures, unless specified otherwise data shown are Mean ± SEM. Different letters indicate significant difference at $p<0.05$, one-way ANOVA followed by Tukey's multiple comparison test.

The online version of this article includes the following source data and figure supplement(s) for figure 1:

**Source data 1.** Excel spreadsheet containing quantitative data for *Figure 1*.
**Figure supplement 1.** Morphology of *trpv6* mutant fish.
**Figure supplement 2.** Sequence alignment of the zebrafish Trpv6, human TRPV5, and human TRPV6 pore region.

(*Figure 2—figure supplement 1*). The NaR cell proliferation rate was significantly elevated in the mutant fish in all stages examined (*Figure 2C*). At five dpf, the *trpv6Δ8* mutant fish had 3-time more NaR cells (*Figure 2C*). Essentially same data were obtained with the *trpv6Δ7* fish (*Figure 2D*). GdCl₃, a Trpv6 inhibitor, was used to further test the role of Trpv6. GdCl₃ treatment increased NaR cell proliferation in the wild-type and heterozygous fish, while it did not further increase NaR cell proliferation in the mutant fish (*Figure 2E* and *Figure 2—figure supplement 2*). Ruthenium red, another Trpv6 inhibitor, had similar effects (*Figure 2—figure supplement 2*). Next, Trpv6 and Trpv6D539A were randomly expressed in NaR cells in *trpv6Δ8⁻/⁻; Tg(igfbp5a:GFP)* fish using a Tol2 transposon BAC-mediated genetic mosaic assay (*Liu et al., 2018*). Reintroduction of Trpv6 reversed the quiescence to proliferation transition (*Figure 2F*) and reduced the apical opening size to the control levels (*Figure 2—figure supplement 1*). Trpv6D539A, however, had no such effect (*Figure 2F*). These

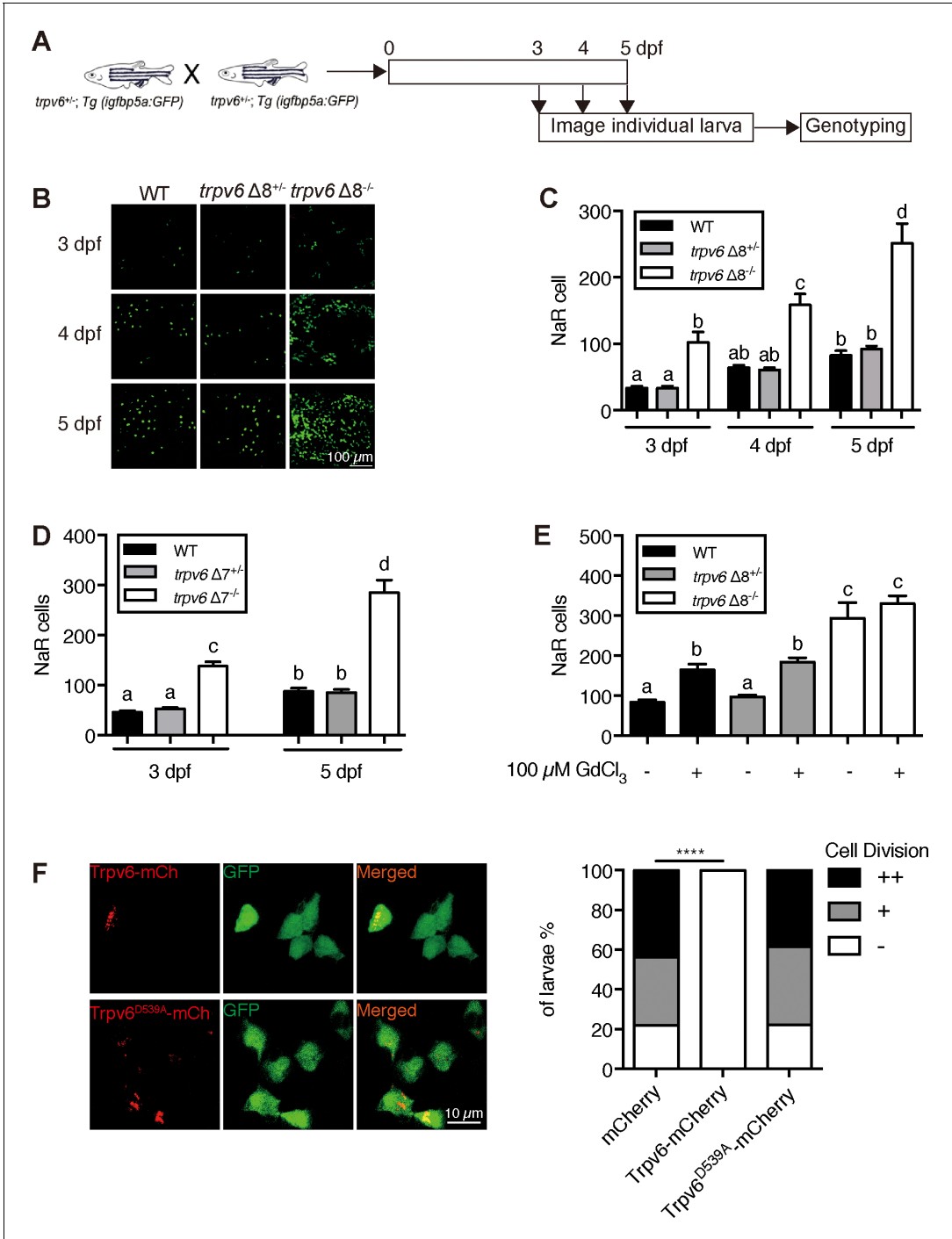

**Figure 2.** Trpv6 regulates epithelial cell quiescence-proliferation decision. (**A**) Diagram of the experimental design. (**B**) Representative images. In this and all subsequent larval images, lateral views of the yolk-sac region are shown with dorsal up and anterior to the left. (**C–D**) Mean NaR cell number/fish of the indicated genotypes. n = 6–9. (**E**) Progenies of *trpv6 Δ8+/-; Tg (igfbp5a:GFP)* intercross were raised to 3 dpf and treated with 100 µM GdCl₃ from 3 to 5 dpf. NaR cells in each fish were quantified following individual genotyping. n = 13–22. (**F**) Progenies of *trpv6Δ8+/-;Tg(igfbp5a:GFP)* intercross were injected with the indicated BAC-mCherry DNA at one-cell stage. At 5 dpf, the Trpv6-expressing NaR cells in each fish were scored following a published scoring system (*Liu et al., 2018*). Representative images are shown in the left and quantified results in the right panel. ****, p<0.0001 by Chi-Square test, fish n = 12–38.

The online version of this article includes the following source data and figure supplement(s) for figure 2:

**Source data 1.** Excel spreadsheet containing quantitative data for *Figure 2*.

**Figure supplement 1.** Genetic deletion of Trpv6 increases epithelial cell apical opening.

*Figure 2 continued on next page*

*Figure 2 continued*

**Figure supplement 2.** Inhibition of Trpv6 increases epithelial cell proliferation.

data showed that Trpv6 functions as a major barrier in the quiescence to proliferation transition and this action requires its $Ca^{2+}$ permeability.

## Trpv6 controls the quiescence-proliferation decision via regulating IGF signaling

Previous studies showed that pre-exiting NaR cells in wild-type fish re-enter the cell cycle in response to low $[Ca^{2+}]$ treatment (*Dai et al., 2014*; *Liu et al., 2017*). To determine whether this effect is related to Trpv6, 3 dpf $trpv6\Delta7^{-/-};Tg(igfbp5a:GFP)$ larvae and siblings were subjected to low $[Ca^{2+}]$

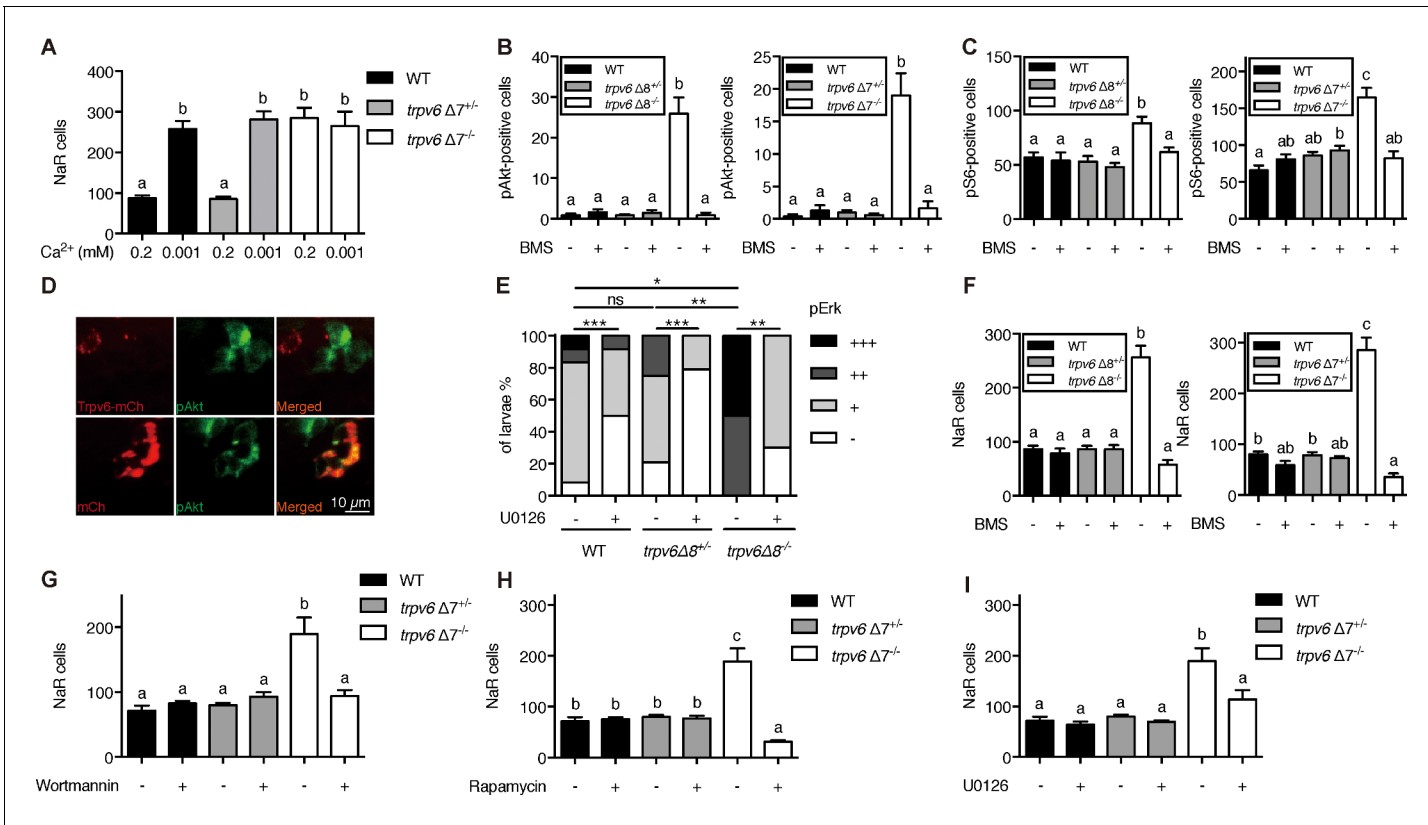

**Figure 3.** Trpv6 prevents the quiescence to proliferation transition via regulating IGF1 receptor-mediated IGF signaling. (**A**) Progenies of $trpv6\Delta7^{+/-}$; *Tg (igfbp5a:GFP)* intercrosses were grown in embryo solutions with the indicated $Ca^{2+}$ concentration from 3 dpf to 5 dpf. NaR cells in each fish were quantified followed by individual genotyping. n = 5–17 fish. (**B–C**) Embryos of the indicated genotypes were raised to 3 dpf and treated with 0.3 μM BMS-754807 or DMSO. At 4 dpf, the treated fish were subjected to immunostaining using an anti-phospho-Akt antibody (**B**) or an anti-phospho-S6 antibody (**C**). Representative images are shown in (*Figure 3—figure supplement 2A and B*). n = 5–41. (**D**) Progenies of a $trpv6\Delta8^{+/-}$; *Tg (igfbp5a:GFP)* intercross were injected with the indicated BAC-mCherry DNA at one-cell stage. At 4 dpf, the larvae were subjected to phospho-Akt and mCherry double staining. (**E**) Embryos of the indicated genotypes were raised to 3 dpf and treated with 30 μM U0126 or DMSO. At 4 dpf, the treated fish were subjected to immunostaining using an anti-phospho-Erk antibody. pErk signals were scaled as shown in *Dai et al. (2014)*. n = 4–24. (**F–I**) Progenies of $trpv6\Delta8^{+/-}$; *Tg(igfbp5a:GFP)* or $trpv6\Delta7^{+/-}$; *Tg(igfbp5a:GFP)* intercrosses were raised to 3 dpf and treated with BMS-754807 (0. 3 μM), Wortmannin (0. 06 μM), Rapamycin (1 μM), U0126 (10 μM) or DMSO from 3 to 5 dpf. NaR cells in each fish were quantified followed by individual genotyping, n = 6–22. The online version of this article includes the following source data and figure supplement(s) for figure 3:

**Source data 1.** Excel spreadsheet containing quantitative data for *Figure 3*.
**Figure supplement 1.** *trpv6, igf1ra, igf1rb* expression in $trpv6^{-/-}$.
**Figure supplement 2.** Akt-Tor pathway activation in $trpv6^{-/-}$.
**Figure supplement 3.** Inhibition of Trpv6 increases Akt signaling.
**Figure supplement 4.** pErk level was elevated in in $trpv6^{-/-}$.

challenge test. Low [Ca$^{2+}$] treatment resulted in a three-fold increase in proliferating NaR cells in the wild-type and heterozygous fish (*Figure 3A*). This value was comparable to that of *trpv6Δ7$^{-/-}$* larvae kept in normal [Ca$^{2+}$] media (*Figure 3A*). Low [Ca$^{2+}$] treatment did not further increase NaR cell number in the mutant larvae (*Figure 3A*). Low [Ca$^{2+}$] treatment significantly increased *trpv6* mRNA level in wild-type fish and heterozygous fish, an adaptive response in Ca$^{2+}$ homeostasis reported previously (*Liu et al., 2017*). This increase, however, was abolished in *trpv6$^{-/-}$* mutant (*Figure 3—figure supplement 1A*), likely due to non-sense mRNA decay of mutant *trpv6* mRNA.

Low [Ca$^{2+}$] stress induces NaR cell proliferation and this has been attributed to the activation of IGF1 receptor-mediated PI3 kinase-Akt-Tor signaling (*Dai et al., 2014*; *Liu et al., 2017*; *Liu et al., 2018*). Gene expression analysis results showed that the *igfr1a* and *igfr1b* mRNA levels were comparable between *trpv6Δ7$^{-/-}$* larvae and siblings (*Figure 3—figure supplement 1B–1C*). Immunostaining results showed significant increases in the number of phosphorylated Akt-positive NaR cells in *trpv6Δ7$^{-/-}$* and *trpv6Δ8$^{-/-}$* larvae kept in the normal [Ca$^{2+}$] embryo medium (*Figure 3B*; *Figure 3—figure supplement 2A*). The levels of phospho-Akt in the siblings were minimal. Blocking Trpv6 channel activity using GdCl$_3$ and Ruthenium red increased phospho-Akt levels in the wild-type fish (*Figure 3—figure supplement 3*). Re-expression of Trpv6 in mutant fish inhibited Akt phosphorylation in NaR cells (*Figure 3D*), indicating that Trp6 is both required and sufficient in suppressing Akt signaling. Tor signaling activity was also significantly elevated in the *trpv6Δ7$^{-/-}$* and *trpv6Δ8$^{-/-}$* mutant larvae (*Figure 3C*; *Figure 3—figure supplement 2B*). Mitogen-activated kinase (MAPK) pathway is another major signaling pathway downstream of the IGF1 receptor (*Duan et al., 2010*). Immunostaining results of pErk signaling activity was significantly increased in *trpv6$^{-/-}$* mutant larvae (*Figure 3E* and *Figure 3—figure supplement 4*). These data show that loss of Trpv6 expression or activity results in elevated IGF signaling in NaR cells.

If Trvp6 regulates the quiescence-proliferation decision by suppressing the IGF1 receptor-mediated signaling, then a blockade of IGF1 receptor and key downstream signaling molecules should inhibit the quiescent to proliferation transition. Indeed, treatment of *trpv6$^{-/-}$* mutant fish with BMS-754807, an IGF1 receptor inhibitor, abolished the quiescence to proliferation transition in mutant larvae (*Figure 3F*). However, IGF1 receptor inhibition did not show any significant effect on NaR cell proliferation in wild-type and heterozygous siblings (*Figure 3F*). Treatment of *trpv6$^{-/-}$* mutant fish with PI3 kinase inhibitor Wortmannin, Tor inhibitor Rapamycin, and Mek inhibitor U0126 had similar effects (*Figure 3G–3I*).

## Trpv6 constitutively conducts Ca$^{2+}$ into epithelial cells and regulates the [Ca$^{2+}$]$_i$ levels in vivo

To investigate Trvp6-mediated Ca$^{2+}$ influx into NaR cells in vivo, we generated the *Tg(igfbp5a: GCaMP7a)* fish, a stable reporter fish line expressing GCaMP7a in NaR cells (*Figure 4—figure supplement 1*). After validating the effectiveness of GCaMP7a in reporting intracellular Ca$^{2+}$ levels ([Ca$^{2+}$]$_i$) (*Figure 4—figure supplement 2*), *trpv6Δ8–2$^{+/-}$*; *Tg(igfbp5a:GCaMP7a)$^{+/-}$* fish were crossed with *trpv6Δ8–2$^{+/-}$* fish and their offspring were screened at three dpf and subsequently genotyped individually. While GCaMP7a-positive cells were observed in ~50% of the siblings as expected, none of the *trpv6Δ8–2$^{-/-}$* mutant larvae had any GCaMP7a-positive cells (*Figure 4A*). Addition of the Ca$^{2+}$ ionophore ionomycin restored GCaMP7a signal in the mutant fish to a level comparable to their siblings (*Figure 4B and C*), thus ruling out the possibility that GCaMP7a expression is altered in the mutant fish. Next, Trpv6 was randomly expressed in NaR cells in the *trpv6 Δ8–2$^{-/-}$*; *Tg(igfbp5a: GCaMP7a)* fish using the genetic mosaic assay. Reinduction of Trpv6 significantly increased GCaMP7a signal levels (*Figure 4D*). Ionomycin treatment did not result in further increase (*Figure 4D*). These genetic and in vivo imaging data argue strongly that Trpv6 is not only critical in conducting Ca$^{2+}$ into NaR cells, but also in maintaining the high [Ca$^{2+}$]$_i$ levels in these cells. We next used Trpv6 inhibitors to block Trpv6 activity in *Tg(igfbp5a:GCaMP7a)* fish. Within 8 min after the GdCl$_3$ treatment, the [Ca$^{2+}$]$_i$ levels became significantly lower and the reduction became more pronounced in 12 and 16 min (*Figure 4E and G*; *Video 1*). When GdCl$_3$ was washed out, the [Ca$^{2+}$]$_i$ levels gradually increased and returned to normal levels after 12 min (*Figure 4F and G*; *Video 2*). Similar results were obtained with Ruthenium red (*Figure 4—figure supplement 3*). Addition of the IGF1 receptor inhibitor BMS-754807 did not change the [Ca$^{2+}$]$_i$ levels in NaR cells (*Figure 4—figure supplement 4*). Therefore, Trpv6 constitutively conducts Ca$^{2+}$ into epithelial cells and continuously maintains high [Ca$^{2+}$]$_i$ levels in vivo.

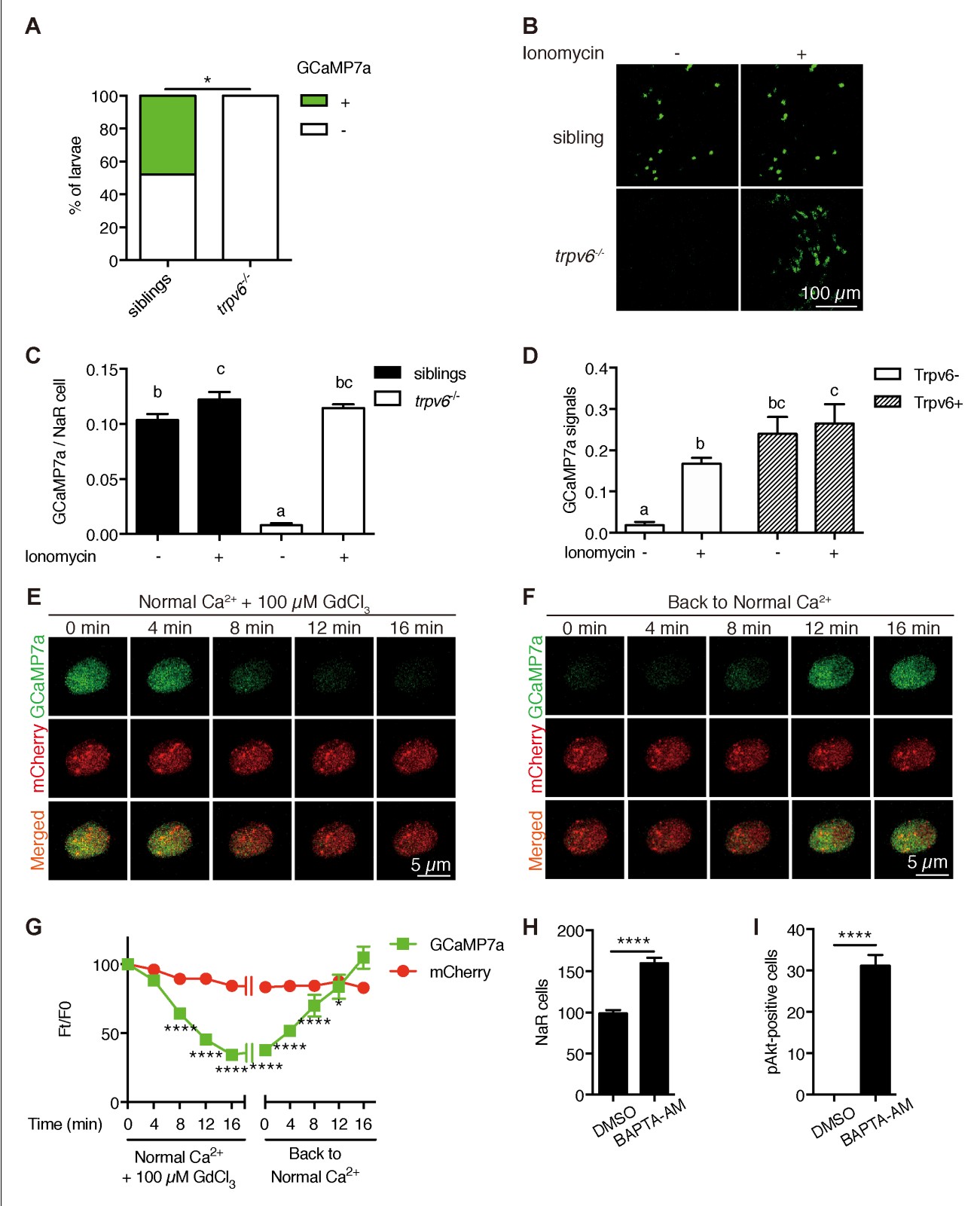

**Figure 4.** Trpv6 is constitutively open and mediates $Ca^{2+}$ influx and maintain high $[Ca^{2+}]_i$ in epithelial cells in vivo. (**A**) *trpv6Δ8–2$^{+/-}$;Tg (igfbp5a: GCaMP7a)$^{+/-}$* was crossed with *trpv6Δ8–2$^{+/-}$*. The progenies were imaged at three dpf followed by individual genotyping. Percentage of GCaMP7a-positive fish is shown. *, p<0.05 by Chi-Square test, n = 21. (**B–C**) Fish described in (**A**) were imaged before and after the addition of 7.5 μM Ionomycin + 10 mM $CaCl_2$. Representative images are shown in (**B**) and the quantified results are shown in (**C**). n = 5–7. (**D**) Progenies from a *trpv6Δ8–2$^{+/-}$;Tg*

*Figure 4 continued on next page*

*Figure 4 continued*

(*igfbp5a:GCaMP7a*)$^{+/-}$ and *trpv6Δ8–2*$^{+/-}$ intercross were injected with *BAC (igfbp5a:Trpv6-mCherry)* DNA at 1 cell stage. They were raised to three dpf. GCaMP7a signal intensity in Trpv6-mCherry-expressing cells and non-expressing NaR cells were quantified before and after the addition of 7.5 μM Ionomycin+10 mM CaCl$_2$. n = 4. (E–G) Time-lapse images of 3 dpf *Tg (igfbp5a:GCaMP7a)* larvae after the addition of 100 μM GdCl$_3$ (E) or following drug removal (F). Changes in GCaMP7a and mCherry signal intensity ratio were quantified and shown in (G). n = 5. * and **** indicate p<0.05 and<0.0001 by Two-way ANOVA followed by Dunnett's multiple comparison test. (H) Wild-type larvae were treated with BAPTA-AM (100 μM) from 3 dpf to 5 dpf. NaR cells were labeled by in situ hybridization using a *trpv6* riboprobe and quantified. (I) Larvae described in (H) were stained for phosphor-Akt after 24 hr treatment. Mean ± SEM. ****, p<0.0001, unpaired t-test. n = 15–19.

The online version of this article includes the following source data and figure supplement(s) for figure 4:

**Source data 1.** Excel spreadsheet containing quantitative data for *Figure 4*.
**Figure supplement 1.** Schematic diagram showing the *BAC(igfbp5a:GCaMP7a)* construct.
**Figure supplement 2.** Validation of *Tg(igfbp5a:GCaMP7a)* fish.
**Figure supplement 3.** Inhibition of Trpv6 decreases [Ca$^{2+}$]$_i$ in NaR cells.
**Figure supplement 4.** Inhibition of IGF1 receptor does not change [Ca$^{2+}$]$_i$ in NaR cells.

## Trpv6 inhibits IGF signaling and epithelial cell proliferation by regulating [Ca$^{2+}$]$_i$ and PP2A is a downstream effector

The observation that [Ca$^{2+}$]$_i$ in NaR cells are continuously maintained at high levels was surprising and intriguing. To determine whether the observed high [Ca$^{2+}$]$_i$ levels have any functional significance, *Tg(igfbp5a:GFP)* larvae were treated with the intracellular Ca$^{2+}$ chelator BAPTA-AM. BAPTA-AM treatment resulted in a significant increase in NaR cell proliferation (*Figure 4H*) and in phospho-Akt signaling levels (*Figure 4I*). These data indicate that the high [Ca$^{2+}$]$_i$ levels are critical in maintaining the quiescent state. To identify the downstream effector(s) of [Ca$^{2+}$]$_i$, a collection of small molecule inhibitors with known protein targets were screened using *Tg(igfbp5a:GFP)* larvae. Okadaic acid and Calyculin A, two inhibitors of the conserved protein phosphatase 2A (PP2A), were among the strongest hits. Treatment of *Tg(igfbp5a:GFP)* larvae with either drug significantly increased NaR cell proliferation (*Figure 5A and B*). This effect is specific because the drug treatment had no such effect in PP2A-deficient zebrafish (*Figure 5—figure supplement 1*). Importantly, the Okadaic acid and Calyculin A treatment-induced NaR cell proliferation was abolished by the IGF1 receptor inhibitor BMS-754807, PI3K inhibitor Wortmannin, Tor inhibitor Rapamycin, and Mek inhibitor U0126 (*Figure 5A–5C*). Okadaic acid or Calyculin A treatment also resulted in significant increases in the phosphorylated-Akt levels in an IGF1 receptor-dependent manner (*Figure 5D*). However, Okadaic acid or Calyculin A treatment did not change the [Ca$^{2+}$]$_i$ levels (*Figure 5—figure supplement 2*), indicating that PP2A acts downstream of the [Ca$^{2+}$]$_i$.

PP2A are a family of conserved protein phosphatases that dephosphorylate Akt and many other proteins (*Perrotti and Neviani, 2013*; *Seshacharyulu et al., 2013*). PP2A holoenzymes are heterotrimers. The core enzyme is made by a catalytic C subunit (Cα and Cβ isoform), a scaffold A subunit (Aα and Aβ), and many regulatory B subunits (*Virshup and Shenolikar, 2009*). The combination of these subunits results in a very large number of different holoenzyme complexes. Our database search suggests that the zebrafish genome contains 3 C subunit genes (*ppp2*ca, cb, and cc). We used CRISPR/Cas9 to transiently knockdown the *ppp2c* genes because stable knockout is likely embryonic lethal. The effectiveness of the targeting guide RNAs was validated (*Figure 5—figure supplement 1*). Transient knockdown of *ppp2cs* resulted in significant increases in the number of proliferating NaR cells (*Figure 5E*) and in phospho-Akt levels

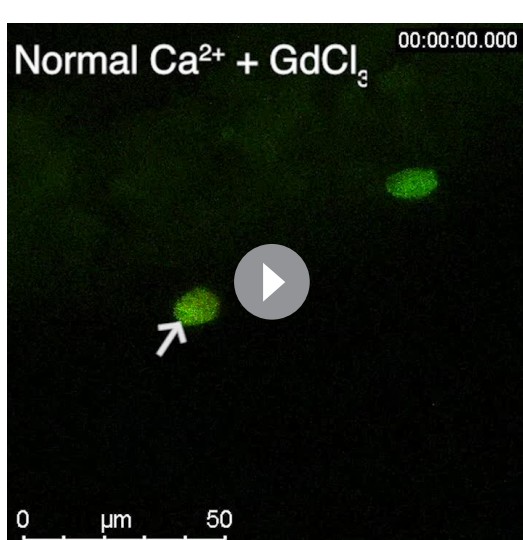

**Video 1.** Normal Ca$^{2+}$ + GdCl$_3$.
https://elifesciences.org/articles/48003#video1

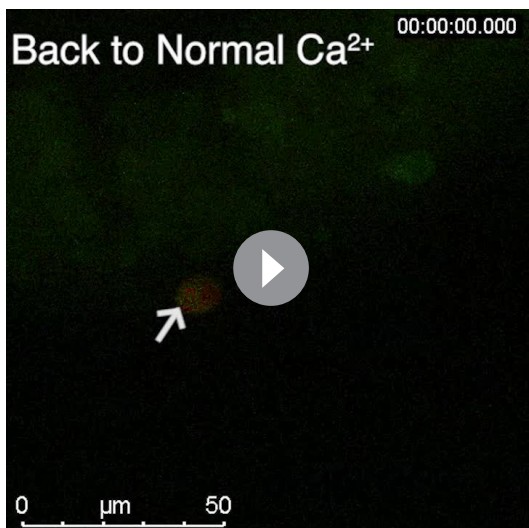

**Video 2.** Back to Normal Ca²⁺.
https://elifesciences.org/articles/48003#video2

(*Figure 5F*), suggesting that PP2A mediates the action of Trpv6-mediated Ca²⁺ influx in zebrafish epithelia.

To determine whether this signaling mechanism is functional in human cells, TRPV6 knockdown experiments were performed in human LoVo colon carcinoma cells using validated siRNA (*Lallet-Daher et al., 2009*). These cells were synchronized by serum starvation followed by serum re-stimulation. Knockdown of TRPV6 resulted in a significant increase in LoVo cell proliferation (*Figure 6A* and *Figure 6—figure supplement 1*). Treatment of LoVo cells with Ruthenium red, GdCl₃, BAPTA-AM, and Okadaic acid all significantly increased cell proliferation (*Figure 6B* and *Figure 6—figure supplement 2*). MTT assay results showed little changes in cell viability in GdCl₃, BAPTA-AM, and Okadaic-acid-treated cells. Ruthenium red treatment resulted in a modest but statistically significant decrease in cell viability (*Figure 6—figure supplement 3*). Finally, LoVo cells were transfected with PP2A-Cα$^{L199P}$ and PP2A-Cα$^{H118N}$, two dominant-negative forms of catalytic subunit Cα of PP2A (*Katsiari et al., 2005*). Expression of PP2A-Cα$^{L199P}$ and PP2A-Cα$^{H118N}$ both significantly increased LoVo cell proliferation (*Figure 6C* and *Figure 6—figure supplement 4*).

## Discussion

In this study, we uncover a previously unrecognized role of Trpv6 and delineates a Trpv6-mediated and evolutionarily conserved Ca²⁺ signaling pathway controlling cell quiescence (*Figure 6D*). We showed that genetic deletion of Trpv6 not only impaired Ca²⁺ uptake and reduced body Ca²⁺ content, but also promoted epithelial cells to exit quiescence and proliferate. Likewise, pharmacological inhibition of Trpv6 increased epithelial cell quiescence-proliferation transition. While low [Ca²⁺] treatment increased epithelial cell proliferation in the siblings, it had no such effect in *trpv6⁻/⁻* larvae, supporting the notion that Trpv6 functions as a major regulator of the quiescent state. Our genetic mosaic analysis results showed that the quiescent state is regulated by Trpv6 in a cell autonomous manner. Reintroduction of Trpv6 in the *trpv6⁻/⁻* mutant fish was sufficient to restore the [Ca²⁺]ᵢ levels, suppress IGF signaling, and reverse the cells back to the quiescent state. The Trpv6D539A mutant had no such activity, suggesting that this action of Trpv6 requires its Ca²⁺ conductance activity.

The impaired Ca²⁺ uptake, reduced body Ca²⁺ content, and premature death observed in *trpv6Δ7* and *trpv6Δ8* mutant fish are in good agreement with a previous study by *Vanoevelen et al. (2011)* using the *mus* mutant fish, but differ considerable from findings made in the mouse model. Bianco et al. reported that *Trpv6* knockout mice were viable, but had reduced intestinal Ca²⁺ uptake, increased urinary Ca²⁺ excretion, decreased bone mineral density, and decreased growth and fertility (*Bianco et al., 2007*). Another *Trpv6⁻/⁻* mutant mouse line reported by Chen et al., however, had normal blood Ca²⁺ concentration and normal bone formation, but with an increased number of osteoclasts (*Chen et al., 2014*). A third *Trpv6⁻/⁻* mutant mouse model showed reduced fertility in male only (*Weissgerber et al., 2012*). The reason(s) of these discrepancies among these mouse studies is not fully understood, but factors such as dietary Ca²⁺ contents may have been critical (*van der Eerden et al., 2012*). Zebrafish embryos continuously lose Ca²⁺ and other ions into the surrounding hypo-osmotic environments and must constantly take up Ca²⁺ from the habitat to survive (*Liu et al., 2018*). This may also contribute to the premature phenotypes found in the zebrafish mutants. Another factor to take into consideration is genetic redundancy. In mammals, there is another closely related TRPV sub-family member, TRPV5. TRPV5 plays similar roles in the transcellular Ca²⁺ transport pathway, although it is mainly expressed in the kidney (*Hoenderop et al., 2005*). It has been shown that TRPV5 expression in the intestine is elevated in *Trpv6* mutant mice

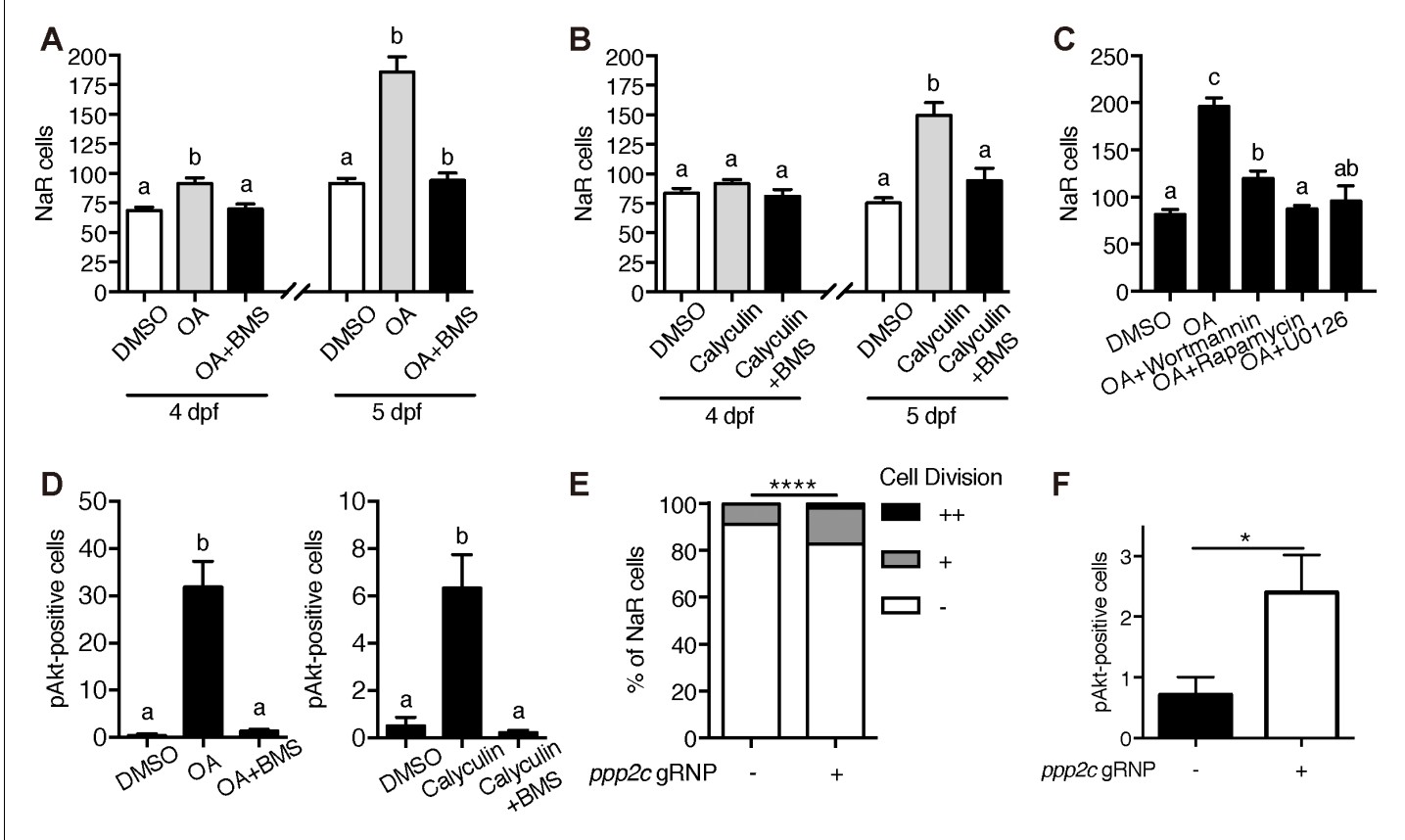

**Figure 5.** PP2A is a downstream effector of Trpv6. (**A–B**) *Tg(igfbp5a:GFP)* embryos were treated with 1 µM Okadaic acid (OA) or 0.1 µM Calyculin A in the presence or absence of 0.3 µM BMS-754807 from 3 dpf. NaR cells were quantified at 4 and 5 dpf. Data shown are n = 10–38. (**C**) *Tg(igfbp5a:GFP)* embryos were treated with 1 µM Okadaic acid (OA) in the presence or absence of Wortmannin (0. 06 µM), Rapamycin (1 µM), U0126 (10 µM) or DMSO from 3 to 5 dpf. NaR cells were quantified at 4 and 5 dpf. Data shown are n = 16–19. (**D**) Wild-type larvae were treated with 1 µM Okadaic acid or 0.1 µM Calyculin A in the presence or absence of 0.3 µM BMS-754807 from 3 dpf to 4 dpf. They were analyzed by immunostaining for phospho-Akt. n = 9–16. (**E**) *Tg(igfbp5a:GFP)* embryos were injected with gRNAs targeting three *ppp2c* genes and Cas9 protein at one-cell stage. They were raised to five dpf. NaR cell division was quantified following a published scoring system (*Liu et al., 2018*). n = 24–28. ****, p<0.0001 by Chi-Square test. (**F**) The embryos treated as in (**E**) were raised to 4 dpf and analyzed by immunostaining for phospho-Akt signal. n = 20–21. *, p<0.05, unpaired t-test.

The online version of this article includes the following source data and figure supplement(s) for figure 5:

**Source data 1.** Excel spreadsheet containing quantitative data for *Figure 5*.
**Figure supplement 1.** Transient knockdown of *pp2a* catalytic subunit genes.
**Figure supplement 2.** Inhibition of PP2A does not change [Ca$^{2+}$]$_i$ in NaR cells.

(*Woudenberg-Vrenken et al., 2012*). In comparison, zebrafish genome has a single trpv6 gene and lacks this genetic redundancy (*Vanoevelen et al., 2011*).

The notion that TRPV6 functions as the primary epithelial Ca$^{2+}$ channel is well supported by findings made in mammalian cells over-expressing TRPV6 (*Fecher-Trost et al., 2017*) and by measuring endogenous TRPV6-mediated Ca$^{2+}$ influx in cultured Jurkat T cells and rat cauda epidermal principle cells (*Gao et al., 2016*; *Kever et al., 2019*). Fura-2 Ca$^{2+}$ imaging studies in cultured mammalian cells transfected with TRPV6 indicated that this channel is constitutively open (*Vennekens et al., 2000*). Although another study failed to detect spontaneous channel activity by patch-clamp recording in cells transfected with TRPV6, a recent study has provided structural and functional evidence that TRPV6 is constitutively active (*McGoldrick et al., 2018*). Other studies have reported that TRPV6 activity is activated by a reduction in [Ca$^{2+}$]$_i$ concentration, and inactivated by higher [Ca$^{2+}$]$_i$ in mammalian cells (*Nilius et al., 2000*; *Yue et al., 2001*). How TRPV6/Trpv6 channel activity is regulated in vivo is less clear. In this study, we generated a reporter fish line using the high-performance genetic calcium reporter GCaMP7a to measure intracellular Ca$^{2+}$ activity in vivo. GCaMPs has been used for

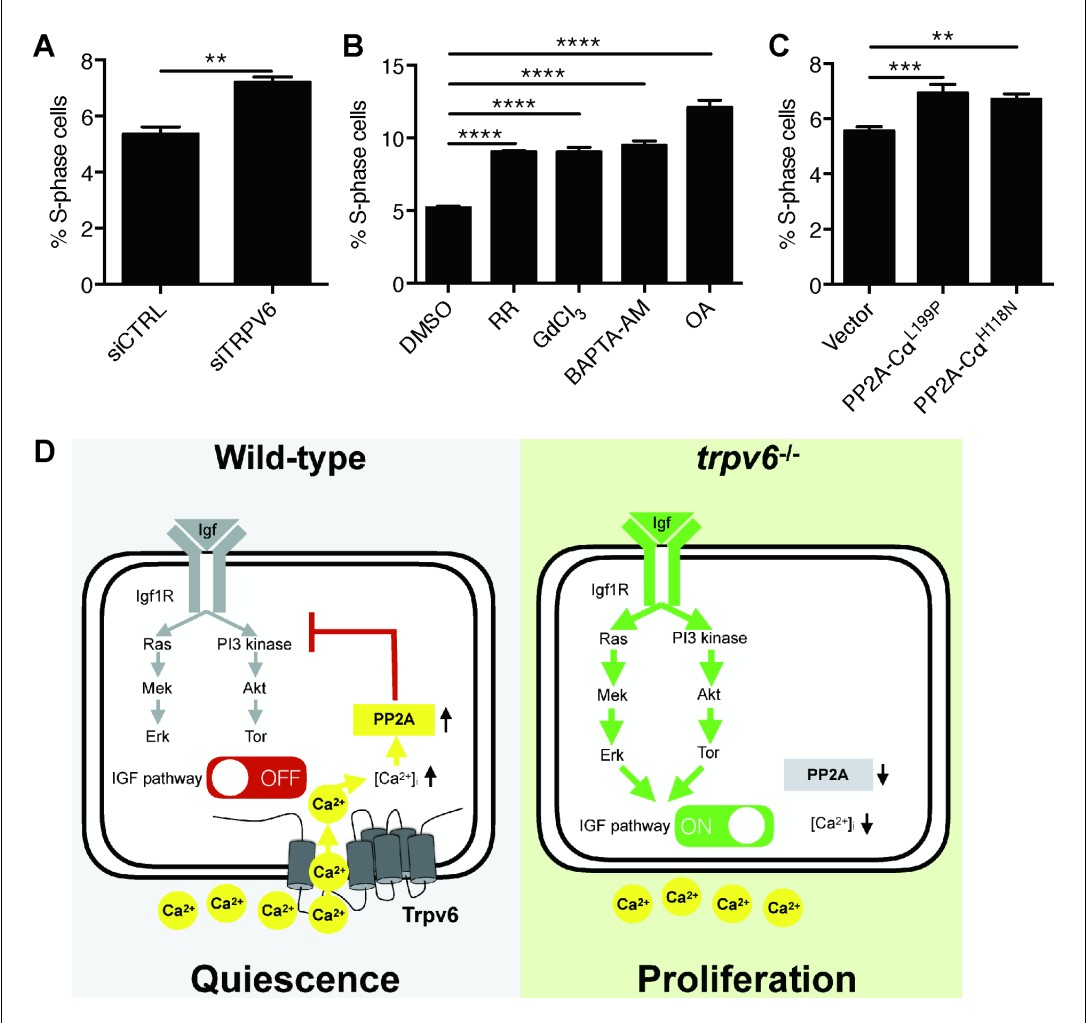

**Figure 6.** Knockdown and inhibition of TRPV6 and PP2A increases human colon carcinoma cell proliferation. (A) LoVo cells transfected with scrambled siRNA or TRPV6 targeting siRNA were synchronized by serum starvation followed by serum re-stimulation. Cells were analyzed by flow cytometry analysis after propidium iodide staining. Percentage of S-phase cells are calculated and shown. Mean ± SEM, n = 3. **, p<0.01 by unpaired t-test. (B) LoVo cells were synchronized by serum starvation. They were re-stimulated with 2% FBS medium containing Ruthenium Red (RR, 100 μM), GdCl$_3$ (100 μM), BAPTA-AM (100 μM), Okadaic acid (OA, 20 nM) or DMSO for 48 hr and analyzed by flow cytometry analysis after propidium iodide staining. Percentage of S-phase cells are shown. Mean ± SEM, n = 3. (C) LoVo cells transfected with the indicated DN-PP2A constructs were synchronized by serum starvation followed by serum re-stimulation. Cells were analyzed by flow cytometry analysis after propidium iodide staining. Percentage of S-phase cells are calculated and shown. Mean ± SEM, n = 3. ****, ***, ** indicate P<0.0001, P<0.001, P<0.01 by one-way ANOVA followed by Tukey's multiple comparison test in (B) and (C). (D) Schematic diagram of the proposed model. See text for details.

The online version of this article includes the following source data and figure supplement(s) for figure 6:

**Source data 1.** Excel spreadsheet containing quantitative data for *Figure 6*.
**Figure supplement 1.** Cell cycle analysis profiles for experiments shown in *Figure 6A*.
**Figure supplement 2.** Cell cycle analysis profiles for inhibitor experiments shown in *Figure 6B*.
**Figure supplement 3.** LoVo cell viability.
**Figure supplement 4.** Cell cycle analysis profiles for experiments shown in *Figure 6C*.

imaging intracellular Ca$^{2+}$ activity in zebrafish neurons (*Muto and Kawakami, 2013*). This approach has alleviated the concern associated with Fura-2 and cell culture systems. Our in vivo Ca$^{2+}$ imaging results showed that the Trpv6 channel is constitutively open in NaR cells in vivo. This conclusion is supported by the facts that genetic deletion of *trpv6* reduced the [Ca$^{2+}$]$_i$ to undetectable levels and re-expression of Trpv6 restored the [Ca$^{2+}$]$_i$ levels in NaR cells. Addition of ionomycin did not result in further increase, indicating the endogenous [Ca$^{2+}$]$_i$ levels are high in these cells. More direct

evidence came from the $[Ca^{2+}]_i$ dynamics analysis results. Within minutes after the addition of Trpv6 blockers $GdCl_3$ or Ruthenium red, the levels of $[Ca^{2+}]_i$ were significantly reduced. When these drugs were washed out, the $[Ca^{2+}]_i$ signal levels returned to normal levels. It was thought that $Ca^{2+}$ transporting epithelial cells, being continuously challenged by $Ca^{2+}$ traffic from the apical side, maintain low levels of $[Ca^{2+}]_i$ using the cytosolic $Ca^{2+}$ binding protein (*Hoenderop et al., 2005*). The in vivo imaging findings made in this study challenge this conventional view. We postulate that maintaining high $[Ca^{2+}]_i$ levels in these $Ca^{2+}$ transporting cells is likely beneficial to the organism because it keeps these cells in differentiated state and functioning as $Ca^{2+}$ transporting units. This idea is supported by the fact that treatment of zebrafish with the intracellular $Ca^{2+}$ chelator BAPTA-AM promoted quiescent epithelial cells to proliferate.

Mechanistically, Trpv6 regulates the quiescent state by mediating constitutive $Ca^{2+}$ influx to continuously down-regulating IGF1-receptor-mediated Akt-Tor and Erk signaling (*Figure 6D*). This is supported by the findings that 1) genetic deletion of Trpv6 or pharmacological inhibition of Trpv6-mediated $Ca^{2+}$ uptake increased Akt-Tor and pErk signaling activity in an IGF1 receptor-dependent manner; 2) re-expression of Trpv6 in the mutant cells suppressed Akt signaling; and 3) inhibition of the IGF1 receptor, PI3 kinase, Tor, and Mek activity abolished or inhibited NaR cell proliferation in $trpv6^{-/-}$ mutant fish. This conclusion is consistent with previous reports that up-regulating of IGF signaling promotes NaR cells to exit the quiescence and re-enter the cell cycle (*Liu et al., 2017*; *Liu et al., 2018*). NaR cells are one of the five types of ionocytes originated from a population of epidermal stem cells, which are specified by the expression of p63 (ΔNp63), a direct target of BMP signaling (*Bakkers et al., 2002*; *Jänicke et al., 2007*; *Lee and Kimelman, 2002*). These epidermal stem cells further develop into Foxi3a/b-activated ionocyte progenitor cells and keratinocyte progenitor cells with low Foxi3a/b expression (*Hsiao et al., 2007*). Krüppel-like factor 4 (Klf4) plays an important role in maintaining the ionocyte progenitor cell pool by stimulating epidermal cell stem proliferation (*Chen et al., 2019*). In addition, several hormones, including isotocin, cortisol and stanniocalcin 1, have been implicated in the regulation of the ionocyte progenitor cell pool (*Chou et al., 2011*; *Chou et al., 2015*; *Yan and Hwang, 2019*). Foxi3a/3b form a positive regulatory loop and this loop is critical in specifying ionocyte progenitors to give rise to the 5 types of ionocytes (*Chang et al., 2009*; *Hsiao et al., 2007*; *Yan and Hwang, 2019*). A hallmark of NaR cells is the expression of Trpv6 (*Liu et al., 2017*; *Pan et al., 2005*). The role of Trpv6 and IGF signaling in NaR cell quiescence regulation unraveled in this study differs from the mechanisms acting in epidermal stem cells and ionocyte progenitor cells, and provides novel insights into the regulation of NaR cell development and function.

Using unbiased chemical biology screens, we have identified PP2A as a key effector of TRPV6/Trvp6 in regulating NaR cell quiescence. Inhibition of PP2A by two distinct inhibitors led to elevated Akt signaling and increased epithelial cell proliferation. Importantly, an IGF1 receptor inhibitor abolished these changes. Likewise, CRISPR/Cas9-mediated transient knockdown of PP2A catalytic subunits increased epithelial cell proliferation and Akt signaling. PP2A can activate Akt and Erk signaling at multiple sites in the IGF signaling pathway (*O'Connor, 2003*). Numerous biochemical studies showing that PP2A can dephosphorylate Akt (*Perrotti and Neviani, 2013*; *Seshacharyulu et al., 2013*). In several human cancer cell lines, PP2A regulates Shc phosphorylation state and upregulates ERK signaling activity in the IGF1-induced signaling pathway (*Yumoto et al., 2006*). Our in vivo findings, together with the in vitro findings indicate that a $[Ca^{2+}]_i$-regulated PP2A isoform(s) likely acts downstream of TRPV6/Trpv6 and regulates the quiescent state in epithelial cells. This is in good agreement with a recent study showing that compromising PP2A activity delays cell cycle exit in *Drosophila* (*Sun and Buttitta, 2015*). The TRPV6-$[Ca^{2+}]_i$-PP2A signaling axis appears to be conserved in human colon carcinoma cells because siRNA-mediated knockdown or inhibition of TRPV6 increased LoVo cell proliferation. Likewise, chelating intracellular $Ca^{2+}$, genetic and pharmacological inhibition of PP2A activity resulted in elevated cell proliferation rate of LoVo cells. The substrate specificity and intracellular distribution of the PP2A holoenzymes are controlled by the B regulatory subunits, which are encoded by a large set of genes classified into four families, that is B, B′, B″, and B‴ (*Virshup and Shenolikar, 2009*). Recent studies suggest that PR72/130 and PR70, 2 members of the B′ family, possess two conserved EF hand motifs (termed EF1 and EF2) and increasing $[Ca^{2+}]_i$ increased the holoenzyme assembly and phosphatase activity (*Kurimchak et al., 2013*; *Magenta et al., 2008*). Another mechanism involves the action of $[Ca^{2+}]_i$-dependent m-calpain: m-calpain degrades PR72 and PR130 into a 45 kDa fragment (*Janssens et al., 2009*). This fragment,

termed as PR45, is resistant to further degradation and exhibits enhanced PP2A activity. These two mechanisms may be related because the calpain-mediated proteolytic activation of PP2A depends on the EF hand integrity (*Janssens et al., 2009*). Future studies are needed to determine whether these mechanisms mediate TRPV6/Trpv6 action in epithelial cells.

Our findings linking the Trpv6-meiated $Ca^{2+}$ uptake to the cellular quiescence regulation have important biomedical implications. Approximately 90% of human cancers arise in epithelial tissues and over-proliferation is one of the cancer hallmarks (*Hanahan and Weinberg, 2011*). The IGF signaling pathway is one of the most frequently mutated signaling pathways in epithelial tissue-derived cancers, including colon and prostate cancers (*Massoner et al., 2010*). The *IGF2* gene and *IRS2* gene are frequently gained in colon cancer (*Cancer Genome Atlas Network, 2012*) and have been proposed as a colorectal cancer 'driver' oncogenes (*Day et al., 2013*). TRPV6 gene is frequently up-regulated in prostate, colon, and other cancer tissues (*Lehen'kyi et al., 2012*; *Prevarskaya et al., 2018*). At present, it is unclear whether the elevated TRPV6 expression promotes tumor growth or it is an adaptive response (*Lehen'kyi et al., 2012*; *Prevarskaya et al., 2018*). Knockdown or overexpression of TRPV6 in cultured cancer cells showed mixed effects in increasing/decreasing proliferation and/or apoptosis (*Chow et al., 2007*; *Lehen'kyi et al., 2007*; *Lehen'kyi et al., 2012*; *Raphaël et al., 2014*; *Skrzypski et al., 2016*). Future studies are needed to clarify the role of TRPV6-$[Ca^{2+}]_i$-PP2A in prostate and colon cancer initiation and progression and its relationship to IGF signaling.

# Materials and methods

## Key resources table

| Reagent type (species) or resource | Designation | Source or reference | Identifiers | Additional information |
|---|---|---|---|---|
| Strain, strain background (*Danio rerio*) | *Tg(igfbp5a:GFP)* | Pubmed ID: 28515443 | ZFIN ID: ZDB-TGCONSTRCT-170830–2 | |
| Strain, strain background (*Danio rerio*) | *trpv6⁻/⁻; Tg (igfbp5a:GFP)* | This paper | | CRISPR/Cas9-mediated knockout |
| Strain, strain background (*Danio rerio*) | *Tg(igfbp5a: GCaMP7a)* | This paper | | Tol2-mediated transgenesis |
| Strain, strain background (*Danio rerio*) | *trpv6⁻/⁻* | This paper | | CRISPR/Cas9-mediated knockout |
| Strain, strain background (*Danio rerio*) | *trpv6⁻/⁻; Tg (igfbp5a:GCaMP7a)* | This paper | | Cross *trpv6⁻/⁻* with *Tg (igfbp5a:GCaMP7a)* |
| Genetic reagent (*Homo sapiens*) | Human TRPV6 siRNA | Pubmed ID: 19270724 | | GACUCUCUAU GACCUCACA |
| Genetic reagent | Mission siRNA Universal Negative Control #1 | Sigma | Catalog no.: SIC001-10nmol | |
| Cell line (*Homo sapiens*) | LoVo | ATCC | RRID:CVCL_039 | |
| Cell line (*Homo sapiens*) | HEK293 | ATCC | RRID:CVCL_0045 | |
| Antibody | Phospho-Akt (Ser473) (Rabbit monoclonal) | Cell Signaling Technology | RRID:AB_2315049 | 1:200 |
| Antibody | Phospho-p44/42 MAPK (Erk1/2) (Thr202/Tyr204) (Rabbit monoclonal) | Cell Signaling Technology | RRID:AB_2315112 | 1:200 |

*Continued on next page*

*Continued*

| Reagent type (species) or resource | Designation | Source or reference | Identifiers | Additional information |
|---|---|---|---|---|
| Antibody | Phospho-S6 Ribosomal Protein (Ser235/236) (Rabbit monoclonal) | Cell Signaling Technology | RRID:AB_2181037 | 1:200 |
| Antibody | Peroxidase-conjugated AffiniPure Donkey Anti-Rabbit IgG (H+L) (Donkey polyclonal) | Jackson ImmunoResearch Laboratories | RRID:AB_10015282 | 1:400 |
| Antibody | Cy3 AffiniPure Goat Anti-Rabbit IgG (H+L) (Goat polyclonal) | Jackson ImmunoResearch Laboratories | RRID:AB_2338006 | 1:300 |
| Antibody | Anti-digoxigenin POD-conjugate (Sheep polyclonal) | Roche | RRID:AB_51450 | 1:500 |
| Recombinant DNA reagent | PP2Ac-L199P | Pubmed ID: 16224536 | | |
| Recombinant DNA reagent | PP2Ac-H118N | Pubmed ID: 16224536 | | |
| Chemical compound, drug | BMS-754807 | Active Biochemicals Co. | Catalog no.: A-1013 | |
| Chemical compound, drug | Wortmannin | Cell Signaling Technology | Catalog no.: 9951 | |
| Chemical compound, drug | Rapamycin | Calbiochem | Catalog no.: 553210 | |
| Chemical compound, drug | U0126 | Cell Signaling Technology | Catalog no.: 9903 | |
| Chemical compound, drug | Okadaic acid | Santa Cruz Biotechnology | Catalog no.: sc3513 | |
| Chemical compound, drug | Calyculin A | Alonmone | Catalog no.: C-100 | |
| Chemical compound, drug | Gadolinium (III) chloride | Sigma-Aldrich | Catalog no.: 439770 | |
| Chemical compound, drug | Ruthenium red | Sigma-Aldrich | Catalog no.: R2751 | |
| Chemical compound, drug | Alizarin red S | Sigma-Aldrich | Catalog no.: A5533 | |
| Chemical compound, drug | Propidium iodide | Sigma-Aldrich | Catalog no.: P4170 | |
| Chemical compound, drug | Fura-2, AM, cell permeant | Invitrogen | Catalog no.: F1221 | |
| Peptide, recombinant protein | Cas9 protein with NLS | PNA Bio | Catalog no.: CP01 | |
| Software, algorithm | GraphPad Prism | | RRID:SCR_002798 | |

## Chemicals and reagents

All chemical reagents were purchased from Fisher Scientific (Pittsburgh, PA) unless stated otherwise. Restriction enzymes were bought from New England Bio Labs (Beverly, MA).

## Zebrafish husbandry

Fish were raised following standard zebrafish husbandry guideline (*Westerfield, 2000*). Embryos were obtained by natural cross and staged following *Kimmel et al. (1995)*. E3 embryo rearing solution (containing 0.33 mM $[Ca^{2+}]$) was prepared as reported (*Westerfield, 2000*). Two additional embryo rearing solutions containing 0.2 mM $[Ca^{2+}]$ (i.e. normal $[Ca^{2+}]$ solution) or 0.001 mM $[Ca^{2+}]$

(i.e. low [Ca$^{2+}$] solution) were made following previously reported formula (*Dai et al., 2014*). To inhibit pigmentation, 0.003% (w/v) N-phenylthiourea (PTU) was added in some experiments. All experiments were conducted in accordance with the guidelines approved by the University of Michigan Institutional Committee on the Use and Care of Animals.

## Generation of *trpv6$^{-/-}$* fish lines using CRISPR/Cas9

The sgRNA targeting *trpv6* (5'-GGGCTCGTTGATGAGCTCCG-3') was designed using CHOPCHOP (http://chopchop.cbu.uib.no/). The sgRNA (30 ng/µl) was mixed with Cas9 protein (700 ng/µl) and co-injected into *Tg(igfbp5a:GFP)* or wild-type embryos at the one-cell stage as described (*Xin and Duan, 2018*). After confirming indels by PCR followed by hetero-duplex assay using a subset of F0 embryos, the remaining F0 embryos were raised to adulthood and crossed with *Tg(igfbp5a:GFP)* or wild-type fish. F1 fish were raised to the adulthood and genotyped. After confirming indels by DNA sequencing, the heterozygous F1 fish were intercrossed to generate F2 fish.

## Transient knockdown of *ppp2cs*

Three sgRNAs targeting *ppp2cs* were designed using CHOPCHOP (http://chopchop.cbu.uib.no/). Their sequences are: *ppp2ca*-sgRNA: 5'-GTTCCATAAGATCGTGAAAC-3'; *ppp2cb*-sgRNA: 5'-GAGCGTTCTCACTTGGTTCT-3'; *ppp2ca2*-sgRNA: 5'-GACGAAGGAGTCGAATGTGC-3'. sgRNAs (30 ng/µl) were mixed with Cas9 protein (700 ng/µl) and co-injected into *Tg(igfbp5a:GFP)* or wild type embryos at the one-cell stage as reported (*Xin and Duan, 2018*). A subset of injected embryos was pooled, DNA isolated, and analyzed by PCR followed by hetero-duplex assays as reported (*Liu et al., 2018*). After confirming the indels, the remaining injected embryos were used for experiments.

## Genotyping

To isolate genomic DNA, pooled embryos or individual adult caudal fin were incubated in 50 µl NaOH (50 mM) at 95℃ for 10 min and neutralized by adding 5 µl 1 M Tris-HCl (pH 8.0). PCR was performed using the following primers: *trpv6*-gt-f, 5'-TGACATTGTGTGTGTTTGTTGC-3'; *trpv6*-gt-r, 5'-GTGAAGGGCTGTTAAACCTGTC-3'; *trpv6*-HMA-f, 5'- GCAGCGGTGGCTTTAATGAAT-3'; *trpv6*-HMA-r, 5'- AAACCTGTCAATCAGAGCACAC-3'; *ppp2ca*-gt-f, 5'- TCACCATCAGTGCATGTCAATA-3'; *ppp2ca*-gt-r, 5'- CTCGATCCACATAGTCTCCCAT-3'; *ppp2cb*-gt-f, 5'- TGGATGATAAAGCGTTTACGAA-3'; *ppp2cb*-gt-r, 5'- ACGTTACACATTGCTTTCATGC-3'; *ppp2ca2*-gt-f, 5'-CTGATGGTTGTGATGCTGTTTT-3'; *ppp2ca2*-gt-r, 5'-CGGTTTCCACAGAGTAATAGCC-3'.

## Morphology analysis

Body length, defined as the curvilinear distance from the head to the end of caudal tail, was measured. Alizarin red staining was performed following a published protocol (*Du et al., 2001*). Images were captured with a stereomicroscope (Leica MZ16F, Leica, Wetzlar, Germany) equipped with a QImaging QICAM camera (QImaging, Surrey, BC, Canada).

## Whole-mount in situ hybridization, and immunostaining

For whole mount immunostaining or in situ hybridization analysis, zebrafish larvae were fixed in 4% paraformaldehyde, permeabilized in methanol, and analyzed as described previously (*Dai et al., 2014*). For double color in situ hybridization and immunostaining, mCherry mRNA signal was detected using anti-DIG-POD antibody (Roche), followed by Alexa 488 Tyramide Signal Amplification (Invitrogen). After in situ hybridization, the stained larvae were washed in 1X PBST and incubated with phosphorylated-Akt antibody overnight at 4℃ and then stained with a Cy3-conjugated goat anti-rabbit immunoglobulin G antibody (Jackson ImmunoResearch). Fluorescent images were acquired using a Nikon Eclipse E600 Fluorescence Microscope with PMCapture Pro six software.

## Plasmid and BAC constructs

The ORF of zebrafish Trpv6 was amplified by PCR using five dpf zebrafish cDNA as template and cloned into pEGFPN1 using the following primers: BglII-zftrpv6-F, 5'-atatAGATCTcgccaccATGCCACCCGCCATATC-3'; no stop-zftrpv6-ca-SalI-R, 5'- TACCGTCGACcaGAGAAACTTGAAATTggggcaatc-3'; Trpv6D539A was engineered by site-directed mutagenesis using the following

primers: zTrpv6_D539A_f, 5′-GGTCAGATTGCCTTGCCAGTGGA-3′; zTrpv6_D539A_r, 5′- TCCAC TGGCAAGGCAATCTGACC-3′. Human TRPV6 ORF was sub-cloned into pEGFPN1 using the following primers: 5′-atatCTCGAGcgccaccATGGGTTTGTCACTG-3′; 5′- TACCGTCGACcaGATCTGATA TTCC-3′. EGFP sequence in those vectors was replaced by mCherry sequence from pmCherry-C1 vector using the following primers: AgeI-mCherry-F, 5′- caACCGGTCGCCACCATGGTGAG-CAAGGGC-3′; mCherry-NotI-stop-r, 5′- TCGCGGCCGCCTACTTGTACAGCTCGTCC-3′. Wild-type zebrafish Trpv6 and Trpv6D539A tagged with mCherry were then inserted into the *igfbp5aBAC* construct to replace the *igfbp5a* coding sequence from the start codon to the end of the first exon through homologous recombination as reported (*Liu et al., 2017*). The primers are: *igfbp5a-zTrpv56*-f, 5′- GTTTTGCCATTTCAAAGCTGGTGAAATAGGTGTTCTACAGTAGGACGA TGCCACCCGCCATATCTGGTGAA-3′ and igfbp5a-pEGFP-C3-Kan-R, . The resulted BAC DNA was validated by sequencing. The validated BAC DNA and Tol2 mRNA were mixed and injected into 1 cell stage *trpv6⁻/⁻; Tg(igfbp5a:GFP)* embryos. Cells co-expressing mCherry and GFP at five dpf were identified and scored using a reported scoring system (*Liu et al., 2018*). PP2Ac$^{L199P}$ and PP2Ac$^{H118N}$ were kind gifts from Dr. George Tsokos, Harvard Medical School. They were subloned into pIRES-mCherry vector using the following primers: EcoRI-PP2A-f: 5′-ccgGAATTCATGGACGAGAAGGTG TTCAC-3′, BamHI-PP2A-r: 5′-cgcGGATCCTTACAGGAAGTAGTCTGGGG-3′ and used in cell transfection.

## Generation of the *Tg(igfbp5a:GCaMP7a)* fish line

*GCaMP7a* DNA was cloned into pEGFPN1 to replace the EGFP sequence using the following primers: BamHI-HA-F, 5′- cgcggatccATGGCATACCCCTACGACG-3′; GCaMP7a-stop-NotI-R, 5′-atttgcggccgcTTACTTAGCGGTCATCATC-3′. The *BAC(igfbp5a:GCaMP7a)* construct was generated following a published protocol (*Liu et al., 2017*). The following primers were used to amplify the GCaMP7a cassette sequence: *igfbp5a*_GCaMP7a_fw, 5′- GTTTTGCCATTTCAAAGCTGGTGAAA TAGGTGTTCTACAGTAGGACGATGGCATACCCCTACGACGTGCCCGAC −3′ and igfbp5a-pEGFP-C3-kan-R The resulted *BAC(igfbp5a:GCaMP7a)* was validated by sequencing. *BAC(igfbp5a: GCaMP7a)* DNA and *Tol2* mRNAs were mixed and injected into zebrafish embryos at one-cell stage. The F0 embryos were screened at 72 hpf by checking GCaMP7a responses to high or low [Ca$^{2+}$] solutions. GCaMP7a-positive F0 embryos were raised and crossed with wild-type fish to obtain F1 individuals. F2 fish were generated by crossing F1 fish.

## Live GCaMP7a imaging

Zebrafish larvae were anesthetized using normal [Ca$^{2+}$] embryo solution supplemented with 0.168 mg/ml tricaine. They were mounted in 0.3% low-melting agarose gel and immersed in 1 ml normal [Ca$^{2+}$] solution. A Leica TCS SP8 confocal microscope equipped with the HC PL APO 93X/1.30 GLYC was used for imaging and LAS X and Image J were used for image analysis.

## RT-qPCR

Zebrafish larvae raised in E3 embryo rearing solution were transferred to normal or low [Ca$^{2+}$] embryo solution from three dpf. Two days later, caudal fin was clipped for genotyping. The larvae of the same genotype and treatment group were pooled and RNA was isolated. Reverse transcription reaction was performed using M-MLV (Invitrogen) oligo(dT)$_{18}$ oligos primer. qPCR was carried out using SYBR Green (Bio-Rad). Primers for qPCR are: *trpv6*-qPCR-F: 5′- GGACCCTACGTCATTGTGA TAC-3′, *trpv6*-qPCR-R: 5′- GGTACTGCGGAAGTGCTAAG-3′, *igf1ra*-qPCR-F: 5′- CGTACCTCAA TGCCAACAAG-3′, *igf1ra*-qPCR-R: 5′- TAGGGCTGTTCGGCTAATGT-3′, *igf1rb*-qPCR-F: 5′- AAAC TTGGGACCAGGGAACT-3′, *igf1rb*-qPCR-R: 5′- ATCTTCTCCCGCTCCACTTC-3′. *18s*-qPCR-F: 5′-AATCGCATTTGCCATCACCG-3′, *18s*-qPCR-R: 5′- TCACCACCCTCTCAACCTCA-3′.

## Cell culture

Human embryonal kidney cells (HEK293) and human LoVo conlon cancer cells were obtained from American Type Tissue Collection (ATCC). Cell identity was authenticated by short tandem repeat (STR) analysis. Cells used were periodically tested for *Mycoplasma* contamination. HEK293 and LoVo cells were cultured in DMEM or DMEM/F12 supplemented with 10% FBS, penicillin and streptomycin in a humidified-air atmosphere incubator containing 5% CO$_2$.

## Fura-2 imaging and electrophysiology recording

HEK293 cells were plated onto 24-mm cover glass coated with L-polylysine and transfected with 0.3 μg of plasmid DNA using Lipofectamine 2000. Twenty-four hours after the transfection, the cover glass was mounted on an imaging chamber and washed with calcium-free Krebs-Ringer HEPES (KRH) solution (118 mM NaCl, 4.8 mM KCl, 1 mM MgCl$_2$, 5 mM D-glucose, and 10 mM HEPES, pH = 7.4). Fura-2 loading was performed following a published method (*Kovacs et al., 2012*). Successfully transfected cells were chosen by mCherry expression. Their cytosolic Ca$^{2+}$ levels were recorded by an EasyRatio Pro system (PTI) at two different wavelengths (340 nm and 380 nm). The Fura-2 ratio (F340/F380) was used to determine changes in intracellular [Ca$^{2+}$]$_i$. At least 50 cells were measured in each slide. Patch clamp recordings were performed at room temperature (*Weissgerber et al., 2012*). The internal pipette solution contained (in mM): Aspartate-Cs 145, NaCl 8, MgCl$_2$ 2, HEPES 10, EGTA 10, Mg-ATP 2, pH 7.2. Normal external solution contained (in mM): NaCl 135, KCl 6, MgCl$_2$ 1.2, HEPES 10, Glucose 12, pH 7.4, supplemented with 10 mM CaCl$_2$ or 30 mM BaCl$_2$. For measuring Ca$^{2+}$ currents, cells were perfused with the normal external solution at first and then switched to solutions as indicated. The DVF solution contains (in mM): NaCl 150, EDTA 10, HEPES 10, pH 7.4. Solutions with different concentration of Ca$^{2+}$ contains (in mM): NaCl 150, HEPES 10, Glucose 12, pH 7.4, supplemented with 0 to 10 mM CaCl$_2$ as indicated.

## Flowcytometry analysis and MTT assay

Human LoVo cells were washed three times with serum-free medium (SFM) and starved in SFM for 12 hr. The cells were then stimulated with 2% FBS medium, with or without inhibitors. Forty-eight hours later, cell cycle analysis was performed using Attune Acoustic Focusing cytometer (Applied Biosystems, Life Technologies) after propidium iodide staining (*Liu et al., 2018*). For siRNA transfection, 100 pmol siRNA and 1.5 μl Lipofectamine RNAiMAX (Invitrogen) was used in each well in 24-well tissue culture plates. For plasmid transfection, 2 μg plasmid and 2 μl Lipofectamine 3000 (Invitrogen) were used in in each well in 24-well tissue culture plates. Six hrs post transfection, cells were grown in complete media containing 10% FBS. There were washed three times with SFM and synchronized by incubation in SFM for 12 hr. The synchronized cells were stimulated with 2% FBS-containing medium for 48 hr and subjected to cell cycle analysis or MTT assay. For MTT assay, 5 mg/ml MTT (Invitrogen) stock solution was diluted with 2% FBS-containing medium and cells were stained for 4 hr at 37°C before dissolving using DMSO. Absorbance at 540 nm was read by a microplate reader (Tecan).

## Statistical analysis

Values are shown as Mean ± standard error of the mean (SEM). Statistical significance between experimental groups was determined using one-way ANOVA followed by Tukey's multiple comparison test or student t-test. Chi-square test was used to analyze the association between two categorical variables. Statistical significances were accepted at $p < 0.05$ or greater.

# Acknowledgements

We thank Dr. George Tsokos, Harvard Medical School, for providing PP2A reagents. This work was supported by NSF grant IOS-1557850 and IOS-1755268. The funders had no role in study design, data collection and analysis, decision to publish, or preparation of the manuscript.

# Additional information

### Funding

| Funder | Grant reference number | Author |
| --- | --- | --- |
| National Science Foundation | IOS-1557850 | Cunming Duan |
| National Science Foundation | IOS-1755268 | Cunming Duan |

The funders had no role in study design, data collection and interpretation, or the decision to submit the work for publication.

## Author contributions
Yi Xin, Data curation, Formal analysis, Investigation, Writing—original draft, Writing—review and editing; Allison Malick, Heya Batah, Investigation, Writing—review and editing; Meiqin Hu, Formal analysis, Investigation, Writing—review and editing; Chengdong Liu, Resources, Formal analysis, Writing—review and editing; Haoxing Xu, Resources, Methodology, Writing—review and editing; Cunming Duan, Conceptualization, Resources, Formal analysis, Supervision, Funding acquisition, Writing—original draft, Project administration, Writing—review and editing

## Author ORCIDs
Haoxing Xu (iD) http://orcid.org/0000-0003-3561-4654
Cunming Duan (iD) https://orcid.org/0000-0001-6794-2762

## Ethics
Animal experimentation: This study was performed in strict accordance with the recommendations in the Guide for the Care and Use of Laboratory Animals of the National Institutes of Health. All experiments were conducted in accordance with the protocol approved by the University of Michigan Institutional Committee on the Use and Care of Animals (Protocol # PRO00008801).

## Decision letter and Author response
Decision letter https://doi.org/10.7554/eLife.48003.sa1
Author response https://doi.org/10.7554/eLife.48003.sa2

# Additional files
## Supplementary files
• Transparent reporting form

## Data availability
All data generated or analysed during this study are included in the manuscript and supporting files.

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
