## [Decision Letter]

Thank you for submitting your article "Cell-autonomous regulation of epithelial cell quiescence by calcium channel Trpv6" for consideration by *eLife*. Your article has been reviewed by three peer reviewers, one of whom is a member of our Board of Reviewing Editors, and the evaluation has been overseen by Didier Stainier as the Senior Editor. The following individual involved in review of your submission has agreed to reveal their identity: Pung Pung Hwang (Reviewer #2).

The reviewers have discussed the reviews with one another and the Reviewing Editor has drafted this decision to help you prepare a revised submission.

Summary:

In this study, using zebrafish epithelial cells, the authors provide evidence that Trpv6 mediates constitutive epithelial Ca^2+^ influx to maintain the quiescent state. This is in agreement with the notion that TRPV6 is indispensable for epithelial cell Ca^2+^ uptake and maintenance of body Ca^2+^ homeostasis. By using genetic or pharmacological deletion of the TRpv6 channel, it was established that the Ca^2+^ conductance function of TRPV6 serves as a barrier in the quiescence to proliferation transition decision in zebrafish NaK-ATPase-rich ionocytes (NaR cells). Re-introduction of the channel restored quiescence in NaR cells. The study then investigated the mechanisms and proposed that Trpv6-mediated calcium flux acts through suppression of the IGF-1-Akt-Tor signaling pathway via the phosphatase 2A (PP2A). A chemical inhibitor screen was used to identify PP2A), and PP2A inhibition increased NaR cell proliferation as well as the phosphorylation of Akt. Furthermore, an IGF-1R kinase inhibitor reversed both the quiescence-proliferation transition as well as the phosphorylation of Akt caused by the inhibition of PP2A. The authors also suggest that this mechanism is conserved in in human colon cancer cells where Trpv6-mediated calcium influx was suggested to activate the PP2A phosphatase and thereby down regulate IGF-1 and Akt signaling.

Overall, this is a novel and well-presented study that greatly benefits from the lack of genetic redundancy within the TRP family in zebrafish.

The following concerns and suggestions should be addressed.

Essential revisions:

1) The authors should show whether the 2-day long, low [Ca^2+^] treatment affects the expression level of TRPV6 in the NaR cells of wild-type or heterozygous TRPV6 fish.

2) It is stated that "TRPV6-mediated Ca^2+^ influx or current has not been recorded in vivo or even in primary epithelial cells (Fecher-Trost, Wissenbach, and Weissgerber, 2017)." However, constitutively active Ca^2+^ current (endogenous) with characteristics that match the epithelial calcium channel TRPV6 have been previously measured in the epithelial principal cells of the rat epididymis (please see Gao et al., 2016). Endogenous TRPV6 currents have also recently been recorded in Jurkat T cells in another study by Kever et al., 2019.

3) The third paragraph of the Discussion puts a question mark on the constitutive activity of the TRPV6 channel overexpressed in mammalian cells and rather speaks for the store-depletion based activation of TRPV6. But it has already been well documented by different and more recent patch-clamp and calcium imaging studies that TRPV6 is constitutively active and its activation does not depend on the store content. See for example the study of McGoldrick et al., 2018, where constitutive currents were recorded in the human TRPV6 transfected HEK293 cells. The Discussion text should be updated accordingly to cover the up-to-date published information on TRPV6 constitutive activity.

4) Several studies have dissected out the molecular pathways of how epithelial cells (including NaR cells) are differentiated from the stem cells and progenitor cells through the regulations by some transcription factors (Foxi3a/b, Gcm2, Klf4, etc.) (Chen et al., 2019; Hsiao et al., 2007; Chang et al., 2009). Some hormones (isotocin, stanniocalain, etc.) were found to positively or negatively regulate the proliferation and differentiation of the mature NaR cells from stem cells and/or progenitor cells (Chou et al., 2011; Chou et al., 2015). I am wondering how much these pathways participate in the epithelial homeostasis. Is there any possible relationship between these pathways and the present proposed signaling in terms of epithelial homeostasis? Can we look at the epithelial homeostasis without considering any effects from the stem cells and/or progenitor cells?

5) The rationale for linking the TRPV6 phenotype with IGF-1 signaling through the Akt/mTOR pathway in a PP2A-dependent manner is not entirely supported by the data presented on the actions of inhibitors of these pathways.

6) To establish a more direct link, it would be worthwhile to know whether the presence or absence of TRPV6 in WT, heterozygous or mutant fish has an impact on the expression levels of IGF-1R.

7) Activation of mitogen activated kinase pathways (MAPK/Erks) by mitogens or growth factor including IGF-1 is generally associated with cell cycle progression. It is surprising that the activation/status of MAPK pathways was not tested or discussed. Demonstration of Akt or TOR activation is not sufficient to conclude that this pathway (activated by IGF-1) is sufficient to drive transition from quiescence to proliferation.

8) The IGF-1R kinase inhibitor BMS is used in a number of experiments, but it is difficult to appreciate what overall effect it has on fish embryos. In Figure 3C there appears to be no affect in WT embryos while BMS suppresses the increase in pAkt observed in TRP6 mutant embryos. Some inhibition of Akt in WT embryos might be expected if IGF-1R is inhibited in WT embryos. Would it be possible to measure IGF-1R activity (e.g. autophosphorylation) to support the observations in Figure 3? Could other IGF-1R activated pathways (e.g. MAPK as discussed above) be tested to support the conclusions?

9) It is stated that, "Treatment of the mutant fish with BMS- 754807, an IGF1 receptor inhibitor, abolished the quiescence to proliferation transition (Figure 3I- 3J)". It would be advisable to explicitly mention here that "However, IGF1 receptor inhibition in heterozygous or WT TRPV6 fish, did not show any significant effect on the NaR cell proliferation."

10) The role of PP2A in NaR cell proliferation and of PP2A in regulating IGF-1R-dependent Akt activity is supported by the inhibitory effects of Okadaic acid and Calyculin. However, the effects of PP2A inhibition on cell proliferation could be independent of Akt. PP2A has well-documented functions in adhesion/growth factor signaling and Shc phosphorylation leading to MAPK activation that should be considered.

11) The experiments with LOVO cells are important to support a role for TRPV6 in human epithelial cells proliferation. But, more data is required to support the data presented in Figure 5F. This includes cell cycle analysis profiles (G1, S, G2/M) as well as some indication of cell viability, cell number/proliferation in the presence and absence of the inhibitors after 48 hours. Moreover, controls showing the on-target effects of the inhibitors under the specific cell culture conditions are necessary.

---

## [Author Response]

Essential revisions:1) The authors should show whether the 2-day long, low [Ca^2+^] treatment affects the expression level of TRPV6 in the NaR cells of wild-type or heterozygous TRPV6 fish.

Following this suggestion, we performed additional experiments. The results, presented in Figure 3—figure supplement 1A, showed that 2-day low [Ca^2+^] treatment increased Trpv6 mRNA levels in wild-type fish and heterozygous fish, part of an adaptive response in Ca^2+^ homeostasis reported previously (Liu et al., 2017). This increase in *trpv6* mRNA levels was abolished in *trpv6*^-/-^ mutant (Figure 3—figure supplement 1A), likely due to non-sense mRNA-mediated decay of mutant *trpv6* mRNA.

2) It is stated that "TRPV6-mediated Ca^2+^ influx or current has not been recorded in vivo or even in primary epithelial cells (Fecher-Trost, Wissenbach, and Weissgerber, 2017)." However, constitutively active Ca^2+^ current (endogenous) with characteristics that match the epithelial calcium channel TRPV6 have been previously measured in the epithelial principal cells of the rat epididymis (please see Gao et al., 2016). Endogenous TRPV6 currents have also recently been recorded in Jurkat T cells in another study by Kever et al., 2019.

We thank the reviewer(s) for bringing our attention to these publications. In the revised manuscript, we have changed the sentences to: “The notion that TRPV6 functions as the primary epithelial Ca^2+^ channel is well supported by in vitro findings made in mammalian cells over-expressing TRPV6 (Fecher-Trost, Wissenbach, and Weissgerber, 2017) and by measuring endogenous TRPV6-mediated Ca^2+^ influx in cultured Jurkat T cells and rat cauda epidermal principle cells (Kever et al., 2019; Gao et al., 2016)”.

3) The third paragraph of the Discussion puts a question mark on the constitutive activity of the TRPV6 channel overexpressed in mammalian cells and rather speaks for the store-depletion based activation of TRPV6. But it has already been well documented by different and more recent patch-clamp and calcium imaging studies that TRPV6 is constitutively active and its activation does not depend on the store content. See for example the study of McGoldrick et al., 2018, where constitutive currents were recorded in the human TRPV6 transfected HEK293 cells. The Discussion text should be updated accordingly to cover the up-to-date published information on TRPV6 constitutive activity.

We agree that more recent studies showing *TRPV6* is constitutively active. We have changed this part to: “Fura-2 Ca^2+^ imaging studies in cultured mammalian cells transfected with TRPV6 indicated that this channel is constitutively open (Vennekens et al., 2000). Although another study failed to detect spontaneous channel activity by patch-clamp recording in cells transfected with TRPV6, a recent study has provided structural and functional evidence that TRPV6 is constitutively active (McGoldrick et al., 2018)”.

4) Several studies have dissected out the molecular pathways of how epithelial cells (including NaR cells) are differentiated from the stem cells and progenitor cells through the regulations by some transcription factors (Foxi3a/b, Gcm2, Klf4, etc.) (Chen et al., 2019; Hsiao et al., 2007; Chang et al., 2009). Some hormones (isotocin, stanniocalain, etc.) were found to positively or negatively regulate the proliferation and differentiation of the mature NaR cells from stem cells and/or progenitor cells (Chou et al., 2011; Chou et al., 2015). I am wondering how much these pathways participate in the epithelial homeostasis. Is there any possible relationship between these pathways and the present proposed signaling in terms of epithelial homeostasis? Can we look at the epithelial homeostasis without considering any effects from the stem cells and/or progenitor cells?

In the revised manuscript, we have added a paragraph discussing published work on the developmental origin of NaR cells and the roles of p63, Foxi3a/3b, *Gcm2,* Klf4, isotocin, cortisol, and stanniocalcin in regulating epidermal stem cells and ionocyte progenitor cells (Discussion, fourth paragraph). All the above mentioned papers are now cited. We pointed out that the previously reported mechanisms are different from the role of Trpv6-regulated IGF signaling discovered in this study. Trpv6 expression is a hallmark of NaR cells. The Trpv6 acts in NaR cells to maintain cellular quiescent state and prevent NaR cell proliferation (Pan et al., 2005; Dai et al., 2014) (see the aforementioned paragraph). Our findings have added yet another layer of regulation in NaR cell biology.

5) The rationale for linking the TRPV6 phenotype with IGF-1 signaling through the Akt/mTOR pathway in a PP2A-dependent manner is not entirely supported by the data presented on the actions of inhibitors of these pathways.

We have obtained additional data linking the TRPV6 phenotype with IGF signaling. Our data showed that i) Akt and Tor signaling activity was activated in *trpv6*^-/-^ larvae kept in the normal [Ca^2+^] embryo medium, while the levels of phospho-Akt in the siblings were minimal (Figure 3B, 3C; Figure 3—figure supplement 2); ii) Blocking Trpv6 channel activity by GdCl_3_ and Ruthenium red increased phospho-Akt levels and NaR cell proliferation in the wild-type fish (Figure 3—figure supplement 3 and Figure 2—figure supplement 2, iii) inhibition of IGF1 receptor abolished the quiescence to proliferation transition in mutant larvae, while it did not have such effect in the wild-type and heterozygous siblings (Figure 3F); and iv) Treatment of *trpv6*^-/-^ mutant fish with PI3 kinase inhibitor Wortmannin and Tor inhibitor Rapamycin abolished NaR cell proliferation transition in mutant larvae, while they had no effects in the siblings (Figure 3G-3I). These data argue strongly that loss of Trpv6 increases NaR cell proliferation via the activation of IGF -Akt-Tor signaling. Our new data also showed that the MAP kinase pathway is also involved (see details in #7).

6) To establish a more direct link, it would be worthwhile to know whether the presence or absence of TRPV6 in WT, heterozygous or mutant fish has an impact on the expression levels of IGF-1R.

Following this suggestion, we measured the expression levels of IGF1 receptor genes by RT-qPCR. Zebrafish has two IGF1 receptor genes, i*gf1ra* and *igf1rb*, due to a teleost specific genome wide duplication event (Maures and Duan, 2002; Schlueter et al., 2006). The data, shown in Figure 4—figure supplement 1B and 1C, showed that the *igfr1a* and *igfr1b* mRNA levels were comparable between *trpv6^-/-^*larvae andsiblings. These results are consistent with our conclusion that Trpv6 suppresses IGF signaling by mediating Ca^2+^ influx and activating PP2A in NaR cells. It is in good agreement with the data presented in Figure 4—figure supplement 4, showing that inhibition of the IGF1 receptor did not change intracellular [Ca^2+^]_i_ in NaR cells.

7) Activation of mitogen activated kinase pathways (MAPK/Erks) by mitogens or growth factor including IGF-1 is generally associated with cell cycle progression. It is surprising that the activation/status of MAPK pathways was not tested or discussed. Demonstration of Akt or TOR activation is not sufficient to conclude that this pathway (activated by IGF-1) is sufficient to drive transition from quiescence to proliferation.

We thank the reviewers for this suggestion. We have carried out additional experiments to address this concern. The new results, presented in Figure 3E, 3I and Figure 3—figure supplement 4, showed that pErk activity was elevated in *trpv6^-/-^*larvae compared with the siblings. The pErk signal was reduced by adding the Mek inhibitor U0126. Furthermore, U0126 treatment significantly decreased NaR cell proliferation in mutant larvae, while it did not have such effect in wild-type and heterozygous siblings. These new data suggest that the MAPK/Erk pathway is also involved in mediating Trpv6 action.

8) The IGF-1R kinase inhibitor BMS is used in a number of experiments, but it is difficult to appreciate what overall effect it has on fish embryos. In Figure 3C there appears to be no affect in WT embryos while BMS suppresses the increase in pAkt observed in TRP6 mutant embryos. Some inhibition of Akt in WT embryos might be expected if IGF-1R is inhibited in WT embryos. Would it be possible to measure IGF-1R activity (e.g. autophosphorylation) to support the observations in Figure 3? Could other IGF-1R activated pathways (e.g. MAPK as discussed above) be tested to support the conclusions?

We have previously shown that treatment of zebrafish embryos with IGF1 receptor kinase inhibitors resulted in reduced growth and slower developmental delay and so did morpholino-based knockdown and genetic inhibition of IGF1 receptor by a dominant negative approach (Schluter et al., 2007, Cell Death Diff. 14:1095; Kamei et al., 2011, Development, 138: 777-786). The current study focused on the local action of IGF signaling in NaR cells. Previous studies suggested that when zebrafish embryos are grown in embryo media containing normal concentrations of [Ca^2+^], NaR cells are maintained in a quiescent state characterized by very low cell division rate and undetectable Akt-Tor activity. Low [Ca^2+^] stress increases the Akt-Tor activity exclusively in an IGF1 receptor-dependent manner in these cells and this in turn promotes quiescent NaR cells to re-enter the cell cycle (Dai et al., 2014; Liu et al., 2017). While the IGF1 receptor genes are widely expressed in many cell types, IGF binding protein 5a (Igfbp5a) is specifically expressed in NaR cells (Dai et al., 2010; 2014; Liu et al., 2017). Genetic deletion of *igfbp5a* blunted the low Ca^2+^ stress-induce IGF-PI3 kinase-Akt-Tor and NaR cell proliferation. Reintroducing Igfbp5a, but not its ligand binding–deficient mutant, restored NaR cell reactivation (Liu et al., 2018).

In this study, we showed that the pAkt and pErk signaling activity was activated in NaR cells in two independent *trpv6^-/-^*lines (but not in their wild-type and heterozygous siblings) when these fish were kept in embryo media containing normal concentrations of [Ca^2+^] (Figure 3B, 3C, and 3E). Addition of the IGF-1R inhibitor BMS abolished pAkt and pErk signaling in NaR cells in *trpv6^-/-^*fish, while they had no effect in siblings (Figure 3B, 3C). These and other evidence have led to the model that Trpv6 suppresses IGF signaling in NaR cells by mediating Ca^2+^ influx and activating PP2A, which down-regulates IGF signaling in the cytosol (Figure 6F). We agree that it would be ideal to directly measure IGF-1R activity in NaR cells. Unfortunately, available mammalian phospho-IGF1R antibodies did not yield specific signals in zebrafish larvae (Dai et al., 2014). Since there are ~ 50 NaR cells in each larva under basal conditions, co-IP with a total IGF1 receptor antibody and an anti-tyrosine antibody is difficult. Our model is consistent with previous biochemical studies showing that PP2A can activate Akt and Erk signaling at multiple sites in the IGF signaling pathway (O’Conner, 2003). Our conclusion is also supported by the fact that inhibition of the IGF1 receptor and PP2A did not change intracellular [Ca^2+^]_i_ in NaR cells (Figure 4—figure supplement 4; Figure 5—figure supplement 2).

9) It is stated that, "Treatment of the mutant fish with BMS- 754807, an IGF1 receptor inhibitor, abolished the quiescence to proliferation transition (Figure 3I- 3J)". It would be advisable to explicitly mention here that "However, IGF1 receptor inhibition in heterozygous or WT TRPV6 fish, did not show any significant effect on the NaR cell proliferation."

Changed as suggested. Thank you.

10) The role of PP2A in NaR cell proliferation and of PP2A in regulating IGF-1R-dependent Akt activity is supported by the inhibitory effects of Okadaic acid and Calyculin. However, the effects of PP2A inhibition on cell proliferation could be independent of Akt. PP2A has well-documented functions in adhesion/growth factor signaling and Shc phosphorylation leading to MAPK activation that should be considered.

We agree. The new data on the role of pErk indicate that PP2A may down-regulate IGF signaling via Akt-dependent and independent mechanisms (Figure 5C). In the revised manuscript, we have added discussions on the possibilities that PP2A may affect IGF signaling through regulating Shc phosphorylation state and increasing MAPK activity (Discussion, fifth paragraph).

11) The experiments with LOVO cells are important to support a role for TRPV6 in human epithelial cells proliferation. But, more data is required to support the data presented in Figure 5F. This includes cell cycle analysis profiles (G1, S, G2/M) as well as some indication of cell viability, cell number/proliferation in the presence and absence of the inhibitors after 48 hours. Moreover, controls showing the on-target effects of the inhibitors under the specific cell culture conditions are necessary.

We have added several lines of new data in the revised manuscript. We showed i) siRNA knockdown of TRPV6 resulted in a significant increase in LoVo cell proliferation (Figure 6A), and ii) expression of PP2A-Cα^L199P^ and PP2A-Cα^H118N^, two dominant-negative PP2A in LoVo cells significantly increased their proliferation (Figures 6C). Representative FCAS profiles are now included in Figure 5—figure supplement 1, 2, 4). The cell viability was assessed by MTT assays. There was no significant change in cell viability in GdCl_3_, BAPTA-AM, and Okadaic acid treated cells but Ruthenium red treatment resulted in a modest but statistically significant decrease in cell viability (Figure 6—figure supplement 2). These new and genetic data provide further support that the TRPV6-[Ca^2+^]_i_-PP2A signaling mechanism is functional in human epithelial carcinoma cells.